# Probabilistic surrogate modeling of damage equivalent loads on onshore and offshore wind turbines using mixture density networks

Deepali Singh[1], Richard Dwight[1], and Axelle Viré[1]

[1]Faculty of Aerospace Engineering, Delft University of Technology, Kluyverweg 1, 2629HS Delft, The Netherlands

**Correspondence:** Deepali Singh (d.singh-1@tudelft.nl)

**Abstract.** The use of load surrogates in offshore wind turbine site assessment has gained attention as a way to speed up the lengthy and costly siting process. We propose a novel probabilistic approach using mixture density networks to map 10-minute average site conditions to the corresponding load statistics. The probabilistic framework allows for the modeling of the uncertainty in the loads as a response to the stochastic inflow conditions. We train the data-driven model on the OpenFAST simulations of the IEA-10MW-RWT and compare the predictions to the widely used Gaussian process regression. We show that mixture density networks can recover the accurate mean response in all load channels with values for the coefficient of determination ($R^2$) greater than $0.95$ on the test dataset. Mixture density networks completely outperform Gaussian process regression in predicting the quantiles, showing an excellent agreement with the reference. We compare onshore and offshore sites for training to conclude the need for a more extensive training dataset in offshore cases due to the larger feature space and more noise in the data.

## 1 Introduction

The selection of a suitable site for the installation of a wind farm plays an important role in limiting installation, maintenance, and operational costs, as well as in ensuring a safe operating lifetime of the structure. This process of *site assessment* or *site-suitability study* typically involves a thorough analysis of the structural integrity of the wind turbine at locations with different site-specific environmental inputs.

Wind turbine certification consists of a rigorous analysis of various design load cases (DLCs) defined by the International Electrotechnical Commission (IEC 61400-3-1, 2019) using time-domain engineering tools such as OpenFAST (Jonkman, 2013), HAWC2 (Larsen and Hansen, 2007), BHawC (Couturier and Skjoldan, 2018; Skjoldan, 2011), to name a few. This step does not typically include site-specific certification. To determine if the wind turbine can handle the loads at a specific site, further analysis is required. The first step involves simulating the DLCs on the type-certified rotor-nacelle assembly with a reference tower. If the site is found suitable, subsequent steps include designing a site-specific tower and performing a site-specific certification. While industry-standard engineering tools are necessary for certification simulations, the preliminary site analysis can benefit from *data-driven surrogate* models to provide quick load estimates. The main objective of this study is to design surrogates that predict the 10-minute load statistics, for instance, 10-minute damage equivalent loads ($DEL^{ST}$), on a fixed-bottom wind turbine using the 10-minute statistics of the environmental conditions as input. This is done using a

probabilistic approach to also quantify the uncertainty in the loads stemming from stochastic variation in the inflow conditions in the 10-minute window.

Modeling the relationship between the $DEL^{ST}$ and environmental conditions is non-trivial. Fatigue is a multiscale phenomenon that depends on the material composition, composite structure, part dimensions, and dynamic inflow. This makes it challenging to model using low-fidelity physics-based approaches. As a consequence, data-driven surrogate models can be beneficial, as they do not require prior knowledge of the underlying physics and can infer complex relationships from observations alone. Especially in systems where analytical closed-form solutions are intractable or the physical properties cannot be easily modeled (Jiang et al., 2020), data-driven surrogates provide a great advantage. Essentially, a surrogate is a simpler and computationally inexpensive representation of the complex system that emulates the outputs as a function of the inputs. Surrogates are widely used as engineering tools for preliminary design calculations, optimization, or real-time control, where accuracy can be reasonably traded for computational efficiency. The data used by the surrogate as ground truth is often from a computational model but can also consist of real-life measurements.

Site-specific load surrogates are often designed using deterministic data-driven modeling approaches (Section 1.2). For a given training dataset $(\boldsymbol{X}, Y) = \{\boldsymbol{x}^q, y^q\}$, where $q = 1...n$, deterministic models map a set of $K$ input features $\boldsymbol{x} \in \mathbb{R}^K$ to the corresponding output $y \in \mathbb{R}$. However, the assumption of a deterministic relationship between inputs and outputs does not hold in our case. For instance, a single value of 10-minute mean wind speed can correspond to an infinite number of inflow patterns, resulting in an infinite number of $DEL^{ST}$ values with a certain probability distribution conditioned on that wind speed. Probabilistic surrogates allow us to model the complete conditional distribution of the $DEL^{ST}$. However, the probabilistic surrogate modeling of offshore wind turbine loads has not been extensively studied in the literature. The primary objective of this study is to develop a probabilistic data-driven surrogate that maps 10-minute statistics of wind and wave conditions ($\boldsymbol{X} \in \mathbb{R}^6$), to the corresponding 10-minute load statistics including $DEL^{ST}$ and standard deviation ($Y \in \mathbb{R}$). We introduce a novel framework for load surrogate modeling that reduces training costs by eliminating seed repetitions without sacrificing prediction accuracy. This probabilistic modeling approach allows for uncertainty propagation and quantification, enabling informed decision-making. In this study, we compare the performance of a highly flexible machine learning method, the mixture density network (MDN) (Bishop, 1994), with the widely used Bayesian approach of Gaussian process regression (GPR). The surrogates are tested using the onshore and offshore versions of the IEA 10-MW reference wind turbine. The load for training the surrogate are calculated using an open-source, multi-fidelity, multi-physics solver called OpenFAST (NREL, 2022; Jonkman, 2013).

## 1.1 The case for probabilistic reasoning in site-specific load surrogates

During the 10-minute period, wind turbines are subjected to randomly varying inflow turbulence and waves, regarding which the surrogate model has no information. For instance, for a given turbulence spectrum, average wind speed, and average turbulence intensity, there are unlimited variations of inflow turbulence patterns that result in equally varied load responses. OpenFAST takes as an input a frozen turbulence field generated by TurbSim using the Kaimal turbulence spectrum inflow model (Kaimal et al., 1972), consisting of stochastic turbulence patterns created using pseudo-number generators initialized

by random seeds. Thus, repeated simulations with the same input parameters but different seeds result in different values of output quantities, yielding a multi-valued mapping, highlighted in Figure 1. On repeating the simulations with sufficient seeds, the load statistics converge towards a value of statistical moments that characterize a random variable, denoted $(Y \mid \boldsymbol{X} = \boldsymbol{x})$. Furthermore, the mean, variance and skewness of the random variable are functions of the controller actions, wind speed, wave period, and turbulence intensity, among other site conditions. The heterogeneity in the variance of the conditional pdf across different operating conditions is known as *heteroscedasticity*.

Deterministic regression models are generally of the type

$$y = f(\boldsymbol{x}) + \varepsilon, \tag{1}$$

where $f$ is a deterministic function of the input features, or the conditional average of the target, and $\varepsilon \sim \mathcal{N}(0, \sigma)$ is the noise component. This framework can only accommodate the conditional statistics of the target. Therefore, when deterministic models are used, the common practice is to convert the multi-valued problem to a single-valued setting. This is done by averaging the response at each sample point over $n$ unique random seeds to approximate quantities such as $\mathbb{E}(Y|\boldsymbol{X} = \boldsymbol{x})$, $\text{Var}(Y|\boldsymbol{X} = \boldsymbol{x})$ or the quantiles of $(Y|X = x)$ (Meinshausen, 2006; Koenker and Hallock, 2001).

The main drawbacks of this approach are as follows.

- A finite sampling of input loading due to stochastic representations of wind and waves in time-domain simulations introduces an uncertainty in the turbine's load response. Zwick and Muskulus (2015) and Müller and Cheng (2018) show that the recommendation by the IEC61400-1 standard to average over a 60-minute long simulation or six 10-minute long simulations is insufficient to fully converge to the average fatigue loads. Liew and Larsen (2022) similarly concluded that some load channels are more sensitive to the number of seeds, and the average of the tower base moments can be off by around $3 - 4\%$, even with $n = 10$. For training surrogates, it is common to run 60-minute-long simulations and obtain the average response or perform four to ten 10-minute simulations over different random seeds to obtain the average response before training the surrogate (Dimitrov et al., 2018; Dimitrov and Natarajan, 2019; Shaler et al., 2022; Slot et al., 2020). The variability in the response with fewer seeds can be interpreted as *noise*, forcing the surrogate model to interpolate the noise in case of a small dataset or fit the mean of the response when the dataset is large. However, the mean may be wrongly inferred if the response is non-Gaussian. Seed repetitions add a significant computational cost to the data generation phase, especially when dealing with expensive simulations like in the case of floating wind turbines.

- Modern wind turbines are equipped with sophisticated controllers that can result in multi-modal responses in loads. Training the model on the expectation of a multi-modal distribution can misrepresent the actual load pattern, as the average of several correct target samples is not necessarily a meaningful target value.

- Most real-world learning tasks involve data sets with complex patterns of missing features that introduce aleatoric uncertainty. Unlike numerical simulations, seed repetitions cannot be performed on such datasets.

An alternate approach is to use probabilistic regression that models the targets as random variables, $Y : \Omega \to \mathbb{R}$, with an unknown conditional pdf, $(Y|\boldsymbol{X} = \boldsymbol{x})$. Probabilistic models provide a framework for informed decision-making by predict-

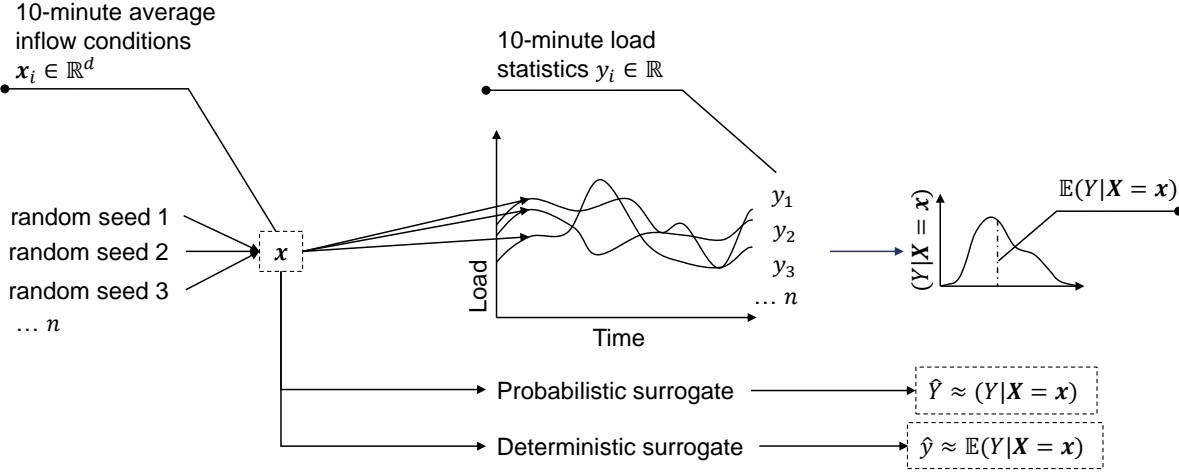

**Figure 1.** Schematic representation of the multi-valued mapping of the input-output pairs used to train the surrogate.

ing a confidence interval in addition to the most likely response. For instance, the uncertainty in the 10-minute $DEL^{ST}$ can be propagated to the lifetime DELs while also considering the distribution of the wind speeds to design less conservative,

site-specific structures. Probabilistic surrogates such as conditional generative adversarial networks and conditional variational autoencoders use latent variables to infer meaningful quantities from data with complex noise distributions (Blei et al., 2017; Yang and Perdikaris, 2019; Kingma and Welling, 2014; Kneib, 2013). Other probabilistic regression approaches like mixture density networks (Bishop, 2006) and generalized lambda distributions (Zhu and Sudret, 2021) use maximum likelihood estimate to infer the conditional distributions in stochastic systems. The probabilistic surrogate modeling approaches can be

trained without having to simulate multiple seed repetitions for each operating condition, as they can infer the conditional response based on neighboring samples. Therefore, training time can be significantly shorter compared to existing frameworks. In addition, it provides estimates of uncertainty with respect to loads and a more robust estimate of mean response. Although it is not the scope of this paper, Bayesian approaches can also be used to differentiate between aleatoric and epistemic sources of uncertainty in the predictions (Olivier et al., 2021; Vega and Todd, 2022; Hlaing et al., 2024). In current design or certification

frameworks, there is no requirement for the quantification of the uncertainty of the loads. However, Li and Zhang (2019) have extrapolated the 50-year accumulated fatigue damage in the mooring lines, tower etc. which could be beneficial for cost reduction when designing the site-specific tower. As we move towards a reliability based design process in the future, probabilistic surrogates will prove to be useful.

## 1.2 Previous work

Wind turbine loads, for site-specific analysis and wind farm design, have commonly been approached with deterministic models like standard artificial neural networks (ANNs), (Schröder et al., 2018; Dimitrov, 2019; Shaler et al., 2022). ANNs are

extremely powerful and emulate the loads well if the training data has been averaged over a set of inflow turbulence realizations. Shaler et al. (2022) compare the performance of inverse distance weighting, ANNs, radial basis functions, Kriging with a partial least squares dimension reduction, and regularized minimal-energy tensor-product b-splines in a wind farm array and observe the highest $R^2$ values for ANNs and the inverse distance weighting method. These approaches, however, do not aim to account for, or predict the variance of the load response.

Kriging, also known as the standard Gaussian process regression (GPR) (Rasmussen and Williams, 2006) is capable of uncertainty quantification but is restricted only to normally distributed homoscedastic responses. Nevertheless, due to its flexibility and ease of implementation, it is widely used as a load emulator to estimate the fatigue load response in wind turbines (Teixeira et al., 2017; Avendaño-Valencia et al., 2021; Li and Zhang, 2019, 2020). Gasparis et al. (2020) compare GPR to other data-driven methods like linear regression and artificial neural networks for modeling power and fatigue loads, showing a superior performance by the GPR. Similarly, Dimitrov et al. (2018) evaluate importance sampling, nearest-neighbor interpolation, polynomial chaos expansion (PCE), GPR, and quadratic response surface (QRS), to conclude a better performance again by the GPR despite a computational penalty. Slot et al. (2020) provide a thorough comparison of the performance of PCE and GPR for the uncertainty quantification of fatigue loads on NREL's 5MW reference onshore wind turbine. They conclude the need for a minimum of four random seeds per training sample in the case of GPR to make high-accuracy predictions. They also note that GPR performs better per invested training simulation than PCE.

Further interest in quantifying the variability of the short-term fatigue loads as a function of the input parameters has initiated research into heteroscedastic surrogates. One of the ways to model heteroscedasticity is through replication-based approaches, wherein the simulations at each set of average input conditions are repeated for multiple realizations of the stochastic field to obtain statistical information about the response. Murcia et al. (2018) use 100 turbulent inflow realizations at each sample point to obtain the first two moments of the fatigue response. Thereafter, they create two independent surrogates using PCE to model the mean and standard deviation of the fatigue loads on the DTU 10MW reference wind turbine. Even though they use only 140 training samples for their model, the replications scale the computational cost by a factor of 100, eventually leading to a very expensive training database. Another replication-based approach is taken by Zhu and Sudret (2020) to model the load response using generalized lambda distributions. In this study, 50 TurbSim realizations are used at each input sample to estimate the four lambda parameters. Four PCE surrogates are then used to model the parameters independently. The main drawback of replication-based methods is the cost of generating the training database, which makes it difficult to apply them to computationally demanding applications such as floating wind turbines. Secondly, the goodness of fit relies heavily on the estimate of the statistical parameters in the first step.

Heteroscedasticity can also be modeled using statistical methods. Abdallah et al. (2019) use parametric hierarchical Kriging to predict blade-root-bending-moment extreme loads that are heteroscedastic on a 2MW onshore wind turbine. Their approach combines low- and high-fidelity observations, where the low-fidelity model informs the high-fidelity GPR. They show that introducing hierarchy helps make the model selection process more robust than the manual tuning of Kriging parameters. Singh et al. (2022) apply chained GPR that uses variational inference within a Bayesian framework to account for heteroscedasticity in the data and make predictions of site-specific load statistics on a more complex case of offshore wind turbines. The model

can capture the heteroscedasticity in a small dataset but is not scalable to high dimensional problems or big data. In order to avoid replication prior to training, Zhu and Sudret (2021) extend the replication-based approach to derive a statistical method combining generalized least-squares with maximum conditional likelihood to estimate the lambda parameters without repli-cations. The main advantage of this method is that it does not assume a Gaussian distribution. However, it can not handle multi-modality.

Only a few approaches attempt to model the uncertainty in the load response of the turbine and the tower, and of those that do, do not consider complex offshore conditions with heteroscedastic multi-modal responses. In this paper, we provide a methodology to build probabilistic data-driven surrogates using MDN (Bishop, 1994). The target is modeled as a mixture of $m \in \mathbb{N}$ Gaussians of varying proportions, capable of generating complex distributions when combined. MDN use feed-forward networks to learn the parameters of the mixture model. The performance of MDN is compared to the standard GPR since it is one of the more widely used and accurate load surrogate modeling approaches in the literature. We train the wind turbine model on stochastic aerodynamic and hydrodynamic features to show the added difficulty in modeling offshore load surrogates.

The layout of the paper is as follows. MDN and GPR are introduced in Section 2. Section 3 presents the setup, including de-tails on the wind turbine model, complex computational model, and the dataset generation methodology. Results are discussed in Section 4, followed by conclusions and future directions in Section 5.

## 2 Regression models

### 2.1 Gaussian process regression

Gaussian process regression (GPR) is a non-parametric, flexible, Bayesian machine learning framework. As mentioned in Section 1.1, the regression problem is defined as,

$$y = f(\boldsymbol{x}) + \varepsilon, \quad \varepsilon \sim \mathcal{N}(0, \sigma^2). \tag{2}$$

The standard GPR models the noise, $\varepsilon$, as a normally distributed quantity with a variance of $\sigma^2$. The function $f(\boldsymbol{x})$ is assigned a Gaussian process prior, that is, $f(\boldsymbol{x}) \sim \mathcal{GP}(\mu(\boldsymbol{x}), k(\boldsymbol{x}, \boldsymbol{x}'))$. The covariance kernel $k$ dictates the smoothness of the function. In this paper, we use the squared exponential kernel defined as,

$$k(\boldsymbol{x}, \boldsymbol{x}') = \sigma_h^2 \exp\left(-\frac{1}{2} \|\boldsymbol{x} - \boldsymbol{x}'\|_{\boldsymbol{l}}^2\right), \quad \|\boldsymbol{x} - \boldsymbol{x}'\|_{\boldsymbol{l}}^2 := \sum_{j=1}^{d} \frac{|x^{(j)} - x'^{(j)}|^2}{l^{(j)}}, \tag{3}$$

implying that the underlying function is smooth and infinitely differentiable, and where $x^{(j)}$ is the $j$-th component of $\boldsymbol{x}$. The characteristic length-scales $\boldsymbol{l} \in \mathbb{R}^d$ are defined per input parameter, and these and the variance $\sigma_h^2$ are *hyperparameters* that are tuned based on the training data. The aim is to make predictions $y^\star$ on unseen data points $\boldsymbol{x}^\star$. The observations $\boldsymbol{y}$ and prediction $y^\star$ are jointly Gaussian, as shown in Equation (4).

$$\begin{bmatrix} \boldsymbol{y} \\ y^\star \end{bmatrix} \sim \mathcal{N}\left(\begin{bmatrix} \mu(X) \\ \mu(\boldsymbol{x}^\star) \end{bmatrix}, \begin{bmatrix} K_{XX} + \sigma^2 I & K_{X\boldsymbol{x}^\star} \\ K_{\boldsymbol{x}^\star X} & K_{\boldsymbol{x}^\star \boldsymbol{x}^\star} + \sigma^2 I \end{bmatrix}\right) \tag{4}$$

The joint distribution is conditioned on the observed values to get the predictive distribution corresponding to a new input $\boldsymbol{x}^\star$ as,

$$y^\star \mid \boldsymbol{y}, X, \boldsymbol{x}^\star \sim \mathcal{N}(\hat{\mu}(x^\star), \hat{\Sigma}(x^\star)) \tag{5}$$

$$\hat{\mu}(\boldsymbol{x}^\star) = \mu(\boldsymbol{x}^\star) + K_{\boldsymbol{x}^\star X}(K_{XX} + \sigma^2 I)^{-1}(\boldsymbol{y} - \mu(X)) \tag{6}$$

$$\hat{\Sigma}(\boldsymbol{x}^\star) = K_{\boldsymbol{x}^\star \boldsymbol{x}^\star} - K_{\boldsymbol{x}^\star X}(K_{XX} + \sigma^2 I)^{-1} K_{X\boldsymbol{x}^\star} + \sigma^2 I \tag{7}$$

$K_{XX}$ is also known as the covariance matrix on $n$ training samples. Computing the inverse of the dense covariance matrix is expensive, with a computational complexity of $\mathcal{O}(n^3)$ and a memory complexity of $\mathcal{O}(n^2)$ (Rasmussen and Williams, 2006). Due to these computational limitations, GPR models are typically not scaled to very large training datasets. In Section 4, GPR training is, therefore, constrained to a maximum of 2500 samples.

The hyperparameters $\sigma_h$ and $l$ may be fixed by the user, but an optimal value is often inferred from the data using type-II maximum likelihood (Rasmussen and Williams, 2006), wherein the negative log marginal likelihood is minimized with respect to the hyperparameters. The negative log marginal likelihood is defined as,

$$-\log p(\boldsymbol{y}|X, \sigma_h, \boldsymbol{l}) =$$
$$\frac{1}{2}(\boldsymbol{y} - \mu(X))^\top (K_{XX} + \sigma^2 I)^{-1}(\boldsymbol{y} - \mu(X))$$
$$+ \frac{1}{2}\log|K_{XX} + \sigma^2 I| + \frac{n}{2}\log 2\pi \tag{8}$$

The Limited memory Broyden–Fletcher–Goldfarb–Shanno (L-BFGS-B) algorithm (Zhu et al., 1997) is used for optimization.

## 2.2 Mixture density networks

A mixture density network is a probabilistic regression method that combines Gaussian mixture models with artificial neural networks (Bishop, 1994). The conditional distribution of the target is represented by a mixture of Gaussian distributions,

$$p(y \mid \boldsymbol{x}) = \sum_{i=1}^{m} \alpha_i(\boldsymbol{x}) \mathcal{N}(y \mid \boldsymbol{\mu}_i(\boldsymbol{x}), \sigma_i^2(\boldsymbol{x})), \tag{9}$$

where $\alpha_i(\boldsymbol{x})$ are the weights or coefficients assigned to the $i^{\text{th}}$ mixture component, and $\mathcal{N}(y \mid \boldsymbol{\mu}_i(\boldsymbol{x}), \sigma_i^2(\boldsymbol{x}))$ is a Gaussian kernel representing the conditional density of the $i^{\text{th}}$ component of the target distribution, with parameters $\boldsymbol{\mu}_i(\boldsymbol{x})$ and $\sigma_i(\boldsymbol{x})$.

Instead of mapping the inflow features $\boldsymbol{x}$ to the load statistics $y$ directly, the neural network is trained to predict the parameter vector, $\boldsymbol{z} \in \mathbb{R}$ consisting of $\alpha_j, \mu_j$ and $\sigma_j$ for $j = 1...m$.

The mixing coefficients $\alpha_i(\boldsymbol{x})$ must sum up to exactly 1. A *softmax* function is used to handle this constraint,

$$\alpha_i = \frac{\exp z_i^\alpha}{\sum_{j=1}^{m} \exp z_j^\alpha}, \tag{10}$$

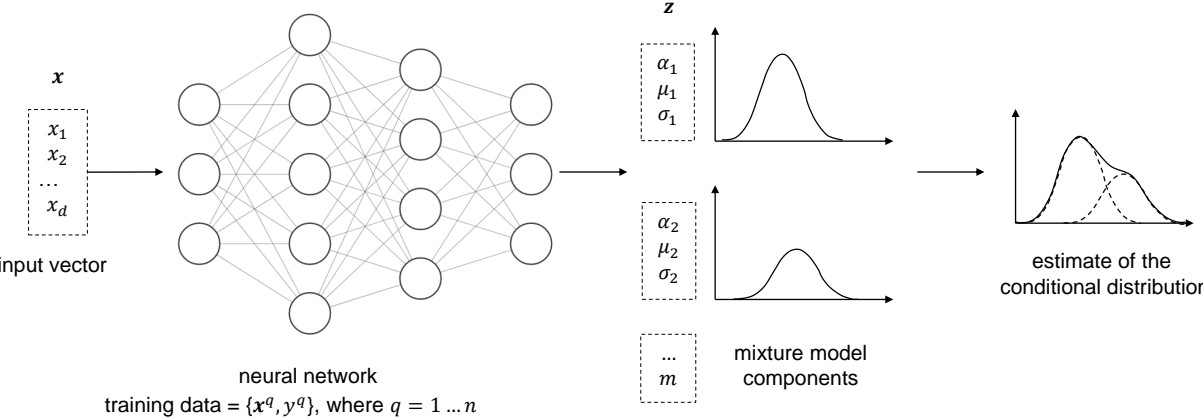

**Figure 2.** Schematic representation of Mixture Density Networks.

where $z_i^\alpha$ are the network outputs predicting the mixture coefficients. Similarly, positive values of the standard deviation are ensured by representing them as exponential functions of the corresponding network outputs, $z_i^\sigma$,

$$\sigma_i = \exp(z_i^\sigma). \tag{11}$$

The likelihood $\mathcal{L}$ of the dataset is given by,

$$\mathcal{L} = \prod_{q=1}^{n} p(y^q \mid \boldsymbol{x}^q) p(\boldsymbol{x}^q) \tag{12}$$

A very commonly used error functions for probabilistic models is the negative log of the likelihood. From (9) and (12), the error can be written as,

$$E^q = -\ln\{\sum_{i=1}^{m} \alpha_i(\boldsymbol{x}^q)\mathcal{N}(y \mid \boldsymbol{\mu}_i(\boldsymbol{x}^q), \sigma_i^2(\boldsymbol{x}^q))\}, \tag{13}$$

where $p(\boldsymbol{x}^q)$ is not included as it is constant with respect to the parameters or weights. The derivative of the error function is calculated at the output layer and is back-propagated to get its gradient with respect to the network weights. Finally, we

have everything we need to minimize the error function using a gradient descent optimization. In this study, we use the Adam optimizer (Kingma and Ba, 2017) to perform stochastic gradient descent. A 10-fold cross-validation set over 600 samples is performed at every training.

The hidden layers in our network use the rectified linear unit (ReLU), defined as,

$$ReLU(x) = \begin{cases} x & \text{for } x > 0 \\ 0 & \text{for } x \leq 0 \end{cases} \tag{14}$$

The output layer of the network does not have an activation function; therefore, the outputs are just linear combinations of the inputs from the previous layer.

Minimizing the error function is an ill-posed problem as there is a conflict between learning the function that fits the data perfectly and remaining robust under varying sets of training data. As the network size grows, the function space increases, and the tendency of the neural network is to overfit. Among several ways to avoid overfitting (Montavon et al., 2012), in this study, we implemented a combination of *early-stopping* (Yao et al., 2007) and *L1* and *L2 regularization* (Ng, 2004).

*Early stopping*

The error function measured on the cross-validation dataset first decreases, then starts increasing as the network begins over-fitting the training data. This can be avoided by applying an early stopping mechanism that stops training as the validation loss stops decreasing over a certain number of iterations. We experimented with a range of early stopping iterations and found 100 to be sufficient for the negative log-likelihood on the validation dataset to converge, but not over-fit. That is, if the validation loss did not show any improvement after 100 iterations, we stopped training the model.

*L1 and L2 regularization*

L1-regularization penalizes the error function with the sum of the magnitude of the weights,

$$E_R^q = E^q + \lambda \sum |w_i| \tag{15}$$

It pushes the coefficients of uninformative features towards zero, effectively pruning the feature space (Tibshirani, 1996).

Weight-decay or L2-regularization, on the other hand, penalizes the error function with a fraction of the squared magnitude of the weights,

$$E_R^q = E^q + \lambda \sum w_i^2 \tag{16}$$

L2-regularization encourages the weights to be small. In both approaches, $\lambda$ is the *regularization parameter*. It is a hyper-parameter that controls the complexity of the model, and the optimal value can be chosen using a search algorithm. We found that heavy regularization with $\lambda = 0.1$ for L1 and L2-regularization deteriorated the results by over-smoothing the conditional response. On the other hand, no L2- regularization also resulted in relatively smaller $R^2$ values for the standard deviation

prediction, likely due to some degree of over-fitting. On the basis of this hyperparameter study on one channel, we decided on
a conservative $\lambda$ value of $1e-3$ for both L1 and L2-regularization.

The main hyperparameters used in this study to train the models to obtain the results in Section 4 are summarized in
Table 1.In subsequent sections, we test the performance of the MDN model with two two-layer network architectures. The first
with a width of 10 nodes per layer, and the second with 30 nodes per layer. The features and targets are scaled with the standard
scaler before training.

**Table 1.** Summary of the network hyperparameters

| Network hyperparameter | Value |
| --- | --- |
| Number of mixture components | 4 |
| Hidden layers | 2 |
| Network width {layer1, layer2} | {10, 10} and {30, 30} |
| Activation function (hidden layers) | ReLU |
| Activation function (output layer) | None |
| Learning rate | 0.005 |
| Maximum epochs | 1000 |
| Mini-batch size | 100 |
| Optimizer | Adam |
| Regularization | |
| $\lambda$ for $L1-$regularization | $10^{-3}$ |
| $\lambda$ for $L2-$regularization | $10^{-3}$ |
| Early-stopping | |
| Early-stopping patience | 100 |
| Early-stopping monitor | validation loss |
| Number of early-stopping validation samples | 600 |

# 3   Setup of the OpenFAST engineering model

## 3.1   OpenFAST modeling approach

The surrogate is modeled on the responses of an aero-hydro-servo-elastic code, OpenFAST, which is used as ground truth in
this study. It is a state-of-the-art, multiphysics numerical tool for modeling wind turbines. It combines analytical and empirical
formulations with conservative assumptions to simplify the code and limit the computational cost. It can model environmental
conditions like stochastic waves, currents, and a frozen wind turbulence field with randomized coherent turbulent structures
superimposed on the random, homogeneous, background turbulence.

OpenFAST is used for setting up a numerical model of the real-world environment and system dynamics to produce the training data for the surrogate. All simulations are performed on the IEA-10MW (Bortolotti et al., 2019) offshore reference wind turbine. TurbSim (Jonkman and Buhl, 2007) simulations for inflow turbulence generation are performed with the Kaimal turbulence spectrum(Kaimal et al., 1972), a grid resolution of 40 points in a $1.16D \times 1.16D$ domain, D being the rotor diameter. The spatial coherence is dictated by the IEC coherence function in the streamwise direction and no coherence in the plane orthogonal to the streamwise direction. The gradient Richardson number is set to 0. The total simulation duration is 900s, out of which the first 300s is discarded to exclude the initial transient. Based on the literature stated in Section 1.1, ten-minute simulations on their own are insufficient for fatigue load estimations. However, the statistical variations in loads that one would expect over longer periods or multiple seed repetitions can be potentially inferred indirectly via probabilistic surrogates based on the variation in the quantities of interest at neighboring training samples. Therefore, ten-minute statistics are sufficient for a complete description of the load response as long as unique turbulence and wave seeds are used for each training simulation.

The loads are calculated with the ElastoDyn module in OpenFAST which uses the Euler-Bernoulli beam theory to calculate the bending moments by assuming the structure to be straight and isotropic. The ServoDyn module is used to control the wind turbine. The controller settings differ from the DTU Wind Energy controller used in the HAWC2 simulations of the IEA-10MW reference document (Bortolotti et al., 2019). The main difference appears at low wind speeds, where the rotor RPM is not restricted to 6, and the collective blade pitch is zero until the rated wind speed. The OpenFAST simulation output with these controller settings at low wind speeds may interfere with the tower's natural frequencies; however, modification of the controller is beyond the scope of this study. The simulations are performed with single precision to limit file size and simulation times without any significant impact on the accuracy of the loads.

The implementation of the IEA-10MW-RWT in OpenFAST is relatively new (Bortolotti et al., 2019). As such, there continue to be constant updates to the public model based on user feedback. This study uses the IEA-10MW-RWT as ground truth for the surrogate modeling study. For that purpose, it need not be perfectly accurate but representative of the expected load response class. The machine learning methodology is expected to be easily transferable to different wind turbine types.

## 3.2 Definition and sampling of the input features

The IEA-10MW-RWT is designed for offshore conditions. However, in this study, we simulate it on both onshore (CASE-ONSHORE) and offshore (CASE-OFFSHORE) sites to be able to evaluate the additional training requirements in the offshore case against an onshore reference.

### 3.2.1 CASE-ONSHORE

Aero-servo-elastic: The wind turbine is subjected only to aerodynamic loading. Wind speed, power-law exponent, and turbulence intensity are selected as the input parameters for the aerodynamic simulations as they have been shown to have the highest impact on the load response in previous studies (Dimitrov et al., 2018). The variable bounds are also the same as the ones defined in (Dimitrov et al., 2018), listed in Table 2. The power-law exponent and turbulence intensity are functions of the wind speed. $R$ and $z$ are the rotor radius and the hub height, respectively. The samples are drawn from a three-dimensional Sobol

sequence (Sobol, 1967) to ensure an even spread of points in the sample space. The random seed for turbulence generation is not included as a training variable. Each sample is therefore associated with a unique random seed, there are no repetitions.

### 3.2.2 CASE-OFFSHORE

Aero-servo-hydro-elastic: The offshore wind turbine is placed on a monopile foundation at 30m water depth and is subject to both aerodynamic and hydrodynamic loading. Along with the aerodynamic parameters of CASE-ONSHORE, there are
290 additional wave parameters in this case as listed in Table 2. For designing load surrogates suitable for multiple sites, ideally the $H_s - T_p$ diagrams from several sites should be combined to define conservative ranges for the two variables. In this study, as an example, we sample the values from a joint $H_s - T_p$ kernel from a representative distribution. Additionally, the minimum and maximum range of $H_s$ may also be defined as a function of wind speed in order to sample from the joint $u - H_s - T_p$ distribution. The first order waves are modeled using the JONSWAP spectrum in HydroDyn. The values of the aerodynamic
variables are the same as in CASE-ONSHORE. In particular, the Turbsim solution files are therefore shared between CASE-ONSHORE and CASE-OFFSHORE. Similar to CASE-ONSHORE, the wave and turbulence seeds are not included in training the model. Each sample is therefore associated with a unique set of random seeds, there are no repetitions.

**Table 2.** List of input variables and the corresponding variable bounds for CASE-ONSHORE (ON) and CASE-OFFSHORE (OFF).

| Case | Variable | Lower Bound | Upper Bound | Sampling |
|------|----------|-------------|-------------|----------|
| ON, OFF | Wind Speed ($u$) [ms$^{-1}$] | 4 | 25 | Uniform, Sobol |
| ON, OFF | Turbulence Intensity ($ti$) [%] | 2.5 | $\frac{18}{u}(6.8 + 0.75u + 3(\frac{10}{u})^2)$ | Uniform, Sobol |
| ON, OFF | Power Law Exponent ($\alpha$) [$-$] | $0.15 - 0.23(\frac{u_{max}}{u})(1 - (0.4\log\frac{R}{z})^2)$ | $0.22 + 0.4(\frac{R}{z})(\frac{u_{max}}{u})$ | Uniform, Sobol |
| OFF | Significant Wave Height ($H_s$) [m] | 0 | 6 | Kernel density estimate, pseudo-random |
| OFF | Spectral Peak Period ($T_p$) [s] | 1 | 21 | Kernel density estimate, pseudo-random |
| OFF | Wave Direction ($wdir$) [deg] | -180 | 180 | Uniform, pseudo-random |
| ON, OFF | Turbulence Random Seed [$-$] | $-50000$ | 50000 | Uniform, pseudo-random |
| OFF | Wave Random Seed 1 [$-$] | $-50000$ | 50000 | Uniform, pseudo-random |
| OFF | Wave Random Seed 2 [$-$] | $-50000$ | 50000 | Uniform, pseudo-random |

### 3.3 Responses

In this paper, we model the 10-minute load statistics at the tower top and tower bottom in the fore-aft direction, and at the
300 blade root in the edgewise and flapwise directions. The statistics include the moment signal's 10-minute standard deviation

and $DEL^{ST}$. The modeled load channels are listed with their notation in Table 3. The model weights for each channel are calibrated separately.

**Table 3.** List of response channels modeled by the data-driven surrogate models.

| Response channel | 10-minute statistics | Notation |
|---|---|---|
| Tower bottom fore-aft moment | stddev, $DEL^{ST}$ | TwrBsMyt |
| Tower top fore-aft moment | stddev, $DEL^{ST}$ | YawBrMyt |
| Blade root edgewise moment | stddev, $DEL^{ST}$ | RootMxb1 |
| Blade root flapwise moment | stddev, $DEL^{ST}$ | RootMyb1 |

Since the direction of the incoming flow is always aligned with the rotor, the fore-aft direction at the tower base is defined in the local coordinate system of the inflow wind. The tower bottom loads must be projected appropriately in the global coordinate system of the wind turbine before integrating to calculate the lifetime fatigue damage in the global coordinate system. In this study, we only calculate the short-term damage in the local coordinate system. The 10-minute fatigue is calculated using $DEL^{ST}$. $DEL^{ST}$ converts the irregular load time series to a constant amplitude and frequency signal that produces an equivalent amount of fatigue damage loads. Rainflow counting (Matsuishi and Endo, 1968) algorithm is used to obtain the load ranges $S_i$ and the number of load cycles $n_i$ needed to calculate the $DEL^{ST}$ as,

$$DEL^{ST} := \left( \frac{n_i S_i^m}{n_{ref}} \right)^{1/m}, \tag{17}$$

where $n_{ref}$ is 600 for 1Hz DELs over 10 minutes. $m$ is the Wöhler coefficient with values 3.5 for the tower, 10 for blade flapwise, and 8 for blade edgewise moments.

### 3.4 Test datasets

The prediction accuracy of the conditional pdf by MDNs is tested on two independently sampled test datasets, that have not been used in the training procedure, referred to as TEST1 and TEST2.

TEST1 consists of 50 pseudo-randomly-selected points spanning the entire sampling domain, with parameter bounds the same as in Table 2. At each test point, engineering simulations with OpenFAST are repeated 300 times to get a reference pdf by keeping the input features constant but changing the turbulence and wave random seeds, resulting in a total of 15000 TurbSim and OpenFAST simulations. In TEST2, we alter only the wind speed and turbulence intensity, keeping the other inflow parameters constant. TEST2 consists of 24 cases, each similarly repeated with 300 turbulence and wave seeds. The purpose of building a separate TEST2 dataset was to be able to visualize the performance of the model through 2-D heat maps in addition to analyzing the integrated quantities derived from TEST1. The values of the parameters are listed in Table 4.

The test and training points for CASE-OFFSHORE are shown in Figure 3. The marginal distributions are shown for the training dataset. The test points for CASE-ONSHORE are identical, but only for the turbulence inflow features, namely, $u$, $ti$, and $\alpha$.

**Table 4.** Variables and their corresponding values in TEST2 dataset. $\Delta$ indicates the discretization step.

| Variable parameters | [Min : Max : $\Delta$] | |
|---|---|---|
| Wind Speed ($u$) [ms$^{-1}$] | [6 : 21 : 3] | |
| Turbulence Intensity ($ti$) [%] | [10 : 40 : 10] | for $u = 6$ |
| | [6 : 24 : 6] | for $u = 9$ |
| | [4 : 16 : 4] | for $u > 10$ |
| Fixed parameters | Value | |
| Power Law Exponent ($\alpha$) [$-$] | 0.08 | |
| Significant Wave Height ($H_s$) [m] | 1 | |
| Spectral Peak Period ($T_p$) [s] | 7 | |
| Wave Direction ($wdir$) [deg] | 0 | |
| Random seeds | Value | |
| Turbulence Random Seed [$-$] | $\mathcal{U}(-50000, 50000)$ | |
| Wave Random Seed 1 [$-$] | $\mathcal{U}(-50000, 50000)$ | |
| Wave Random Seed 2 [$-$] | $\mathcal{U}(-50000, 50000)$ | |

Figure 4 shows, as an example, the 10-minute average tower bottom fore-aft moment as a function of wind speed, along with the conditional distributions at two wind speeds from the TEST1 dataset. The surrogate models aim to predict this kind of conditional variation in the loads due to the stochastic inflow without the need for seed repetitions during training.

### 3.5 Accuracy metric

The qualitative assessment of the performance of the surrogate model is based on two criteria: the coefficient of determination and the Wasserstein distance, as further described hereafter.

### 3.5.1 Coefficient of determination $R^2$

The coefficient of determination, also known as the $R^2$, is a common measure of the goodness of fit of a model. It is defined as,

$$R^2 = 1 - \frac{\sum (y_i - \hat{y}_i)}{\sum (y_i - \bar{y})}, \tag{18}$$

where $\hat{y}_i$ is the predicted output, $y_i$ is the observed value and $\bar{y}$ is the mean of the observed values. $R^2$ is interpreted as the linear correlation between the predicted and observed values of the output vector. To assess the accuracy of the predicted conditional distribution of the response compared to the OpenFAST reference (Figure 4), we calculate the $R^2$ value for the

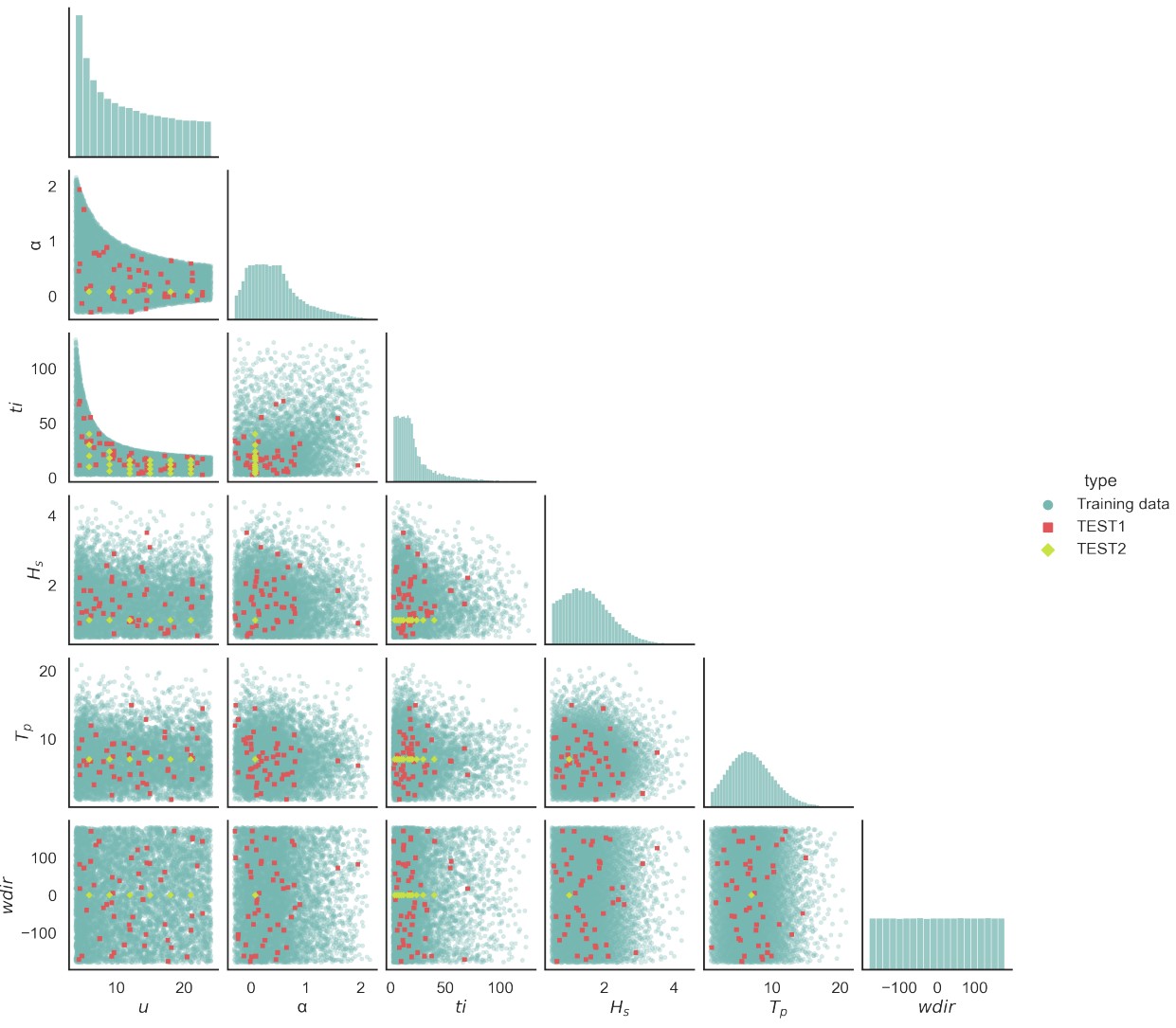

**Figure 3.** Pairplot of the input features for CASE-OFFSHORE showing the training samples along with the test datasets. The marginal distributions are shown along the diagonal for the training dataset.

conditional pdf's mean, standard deviation, $5\%$, and $95\%$ quantiles.These four quantities are derived empirically by obtaining 5000 samples from the surrogate- predicted conditional distribution, and 300 turbulence seed repetitions per test case.

### 3.5.2  Wasserstein distance

The Wasserstein metric is a distance function to compare the difference between the pdfs of any two random variables. It is symmetric, non-negative, and satisfies the triangle inequality, making it a proper distance metric. In the case of 1-D distribu-

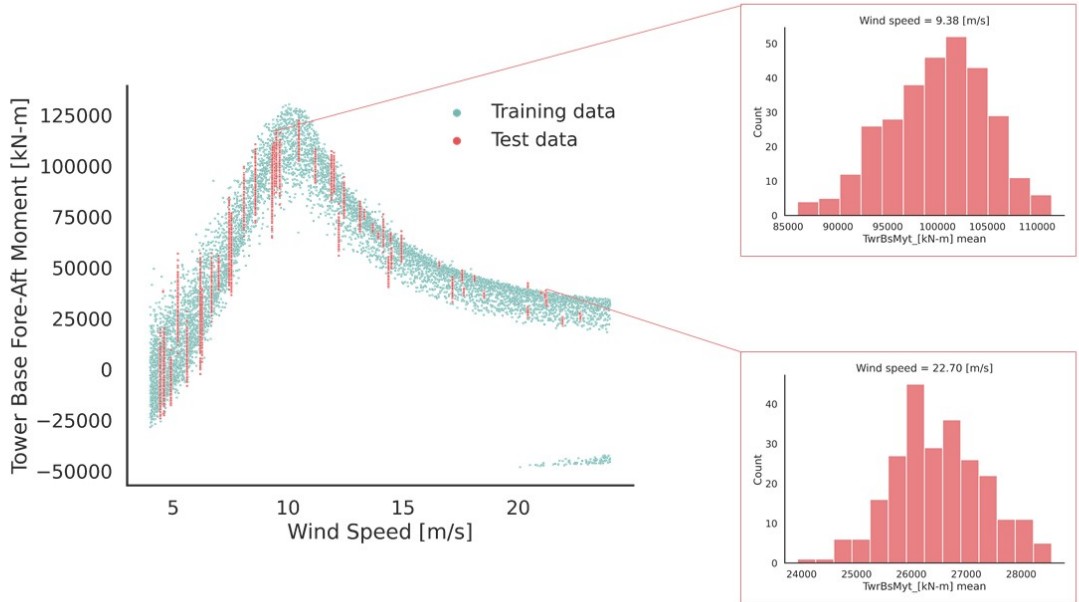

**Figure 4.** The left plot shows the 10-minute average tower bottom fore-aft moment as a function of wind speed. The samples belong to the training database of CASE-OFFSHORE. On the right, two examples of reference histograms generated using 300 turbulence seed repetitions in OpenFAST at wind speeds of $9.38\text{ms}^{-1}$ and $22.7\text{ms}^{-1}$ are shown from the TEST1 dataset.

tions, the Wasserstein-2 distance between a reference empirical measure $Y$ and predicted measure $\hat{Y}$, is defined as (Villani,
2009; Peyré and Cuturi, 2019; Ramdas et al., 2015),

$$W_2(Y,\hat{Y}) = (\int_0^1 |F^{-1}(t) - G^{-1}(t)|^2 dt)^{1/2} \tag{19}$$

where $F^{-1}$ and $G^{-1}$ are the quantile functions of $Y$ and $\hat{Y}$ respectively. The individual quantile functions are obtained from the samples of the empirical distributions and then integrated. In this paper, we calculate the Wasserstein distance between the conditional distribution predicted for each sample ($\hat{Y}$) and the conditional distribution obtained as a reference through seed
repetitions in OpenFAST ($Y$). $\hat{Y}$ consists of 5000 samples, and $Y$ is obtained from 300 turbulence seed repetitions. The distance metric is normalized by the standard deviation of the reference conditional distribution, $Y$. Therefore, a value of $\frac{W_2}{\sigma(Y)} = 1$ is the distance between a distribution with mean $\mu(Y)$, scale $\sigma(Y)$, and a degenerate distribution with the same mean. We calculate the global performance of the model by averaging the normalized Wasserstein distance over $N_{test}$ test samples as,

$$d_{W2} = \mathbb{E}_{N_{test}} \left( \frac{W_2}{\sigma(Y)} \right) \tag{20}$$

## 4 Results and discussion

In this section, we assess how well the surrogate models predict the conditional load distribution on the TEST1 and TEST2 datasets mentioned in Section 3.4. The first part focuses on convergence studies, specifically the impact of training data size on the prediction of the average 10-minute standard deviation of the tower bottom fore-aft moment. The goal is to measure how the accuracy of the predictions varies based on the hyperparameters and initialization of the optimization algorithm. Once the network architecture and training sample size are fixed, we do a rigorous analysis of the model's performance in Section 4.2.

### 4.1 Convergence

Generally speaking, more data translates to better accuracy. However, an increase in data after a certain point gives diminishing returns in accuracy. Given the computational cost of generating the training database, we want to ensure a good model fit with as little training data as possible.

In this section, we look at the convergence of the model with respect to the number of training samples in two two-layer networks with 10 ($[10, 10]$) and 30 ($[30, 30]$) units in each layer. Two different networks are chosen to comment on the robustness of the approach with respect to the network architecture. The convergence study is performed both on CASE-ONSHORE in Figure 5 and CASE-OFFSHORE in Figure 6. The boxes in Figure 5 and Figure 6 extend between the first ($Q1$) and third ($Q3$) quartile of the data and the horizontal line across the box indicates the median. The difference between $Q1$ and $Q3$ defines the interquartile range ($IQR$). The upper whisker extends to the largest data values that are within $1.5IQR$ above $Q3$. The lower whisker, similarly, extends to the lowest data point within $1.5IQR$ below $Q1$. Outliers are visible as dots beyond the whisker boundaries.

The network is trained on the tower bottom fore-aft moment standard deviation (TwrBsMyt [kN-m] stddev). In principle, the convergence study could be performed on any or all of the load channels mentioned in Section 3.3. However, TwrBsMyt [kN-m] stddev was chosen for because it is found to have relatively the highest average $d_{W2}$ value amongst all channels. Models are data-dependent, and each channel would ideally need a separate sensitivity study in terms of the number of training samples needed for convergence, the model architecture, and the model hyperparameters. However, for practical purposes, we assume that if a model can perform well on the channel with the poorest metrics, then it should also perform well, if not optimally, on the other load channels with the same network architecture. Care is taken to prevent overfitting by comparing the evolution of the negative log-likelihood on the training and test datasets during training. At every $N_{train}$, the model training is repeated on 25 uniquely sampled subsets of the data with 10-fold cross-validation. The plots in Figure 6 show the convergence of the model in terms of predicting the normalized 2-Wasserstein distance and statistics, including the response pdf's mean, $5\%$ quantile and $95\%$ quantile. The metrics are averaged over the validation dataset formed by combining TEST1 and TEST2.

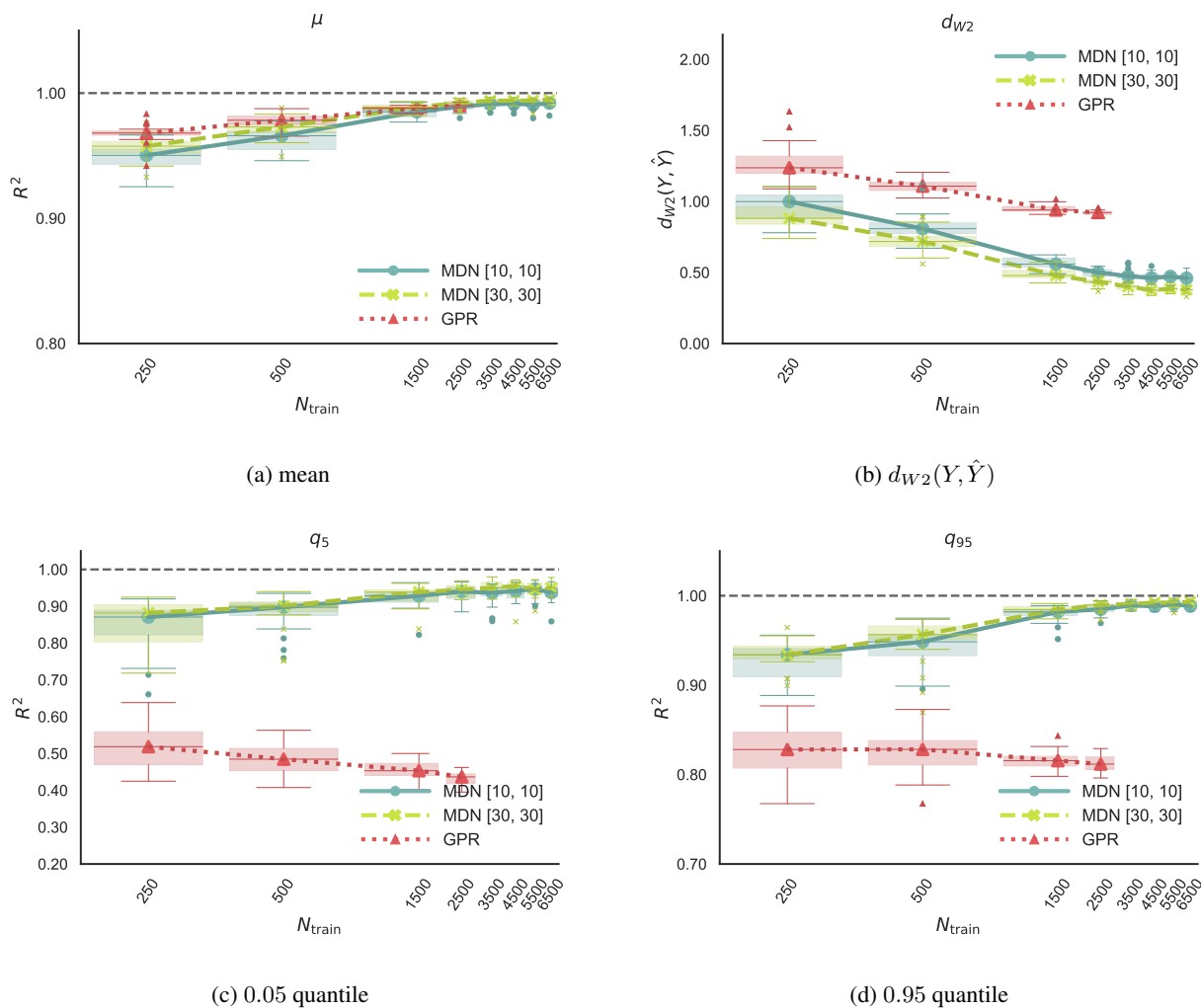

(a) mean

(b) $d_{W2}(Y,\hat{Y})$

(c) 0.05 quantile

(d) 0.95 quantile

**Figure 5.** CASE-ONSHORE: Figures showing the change in the normalized 2-Wasserstein distance, $R^2$ value of the mean, 0.05 quantile and 0.95 quantile of the predicted pdf as a function of the training samples. The study is performed on the tower base fore-aft moment standard deviation (TwrBsMyt stddev). The numbers in the square bracket $[x,x]$ denote the widths of the first and second layer of the 2-layer neural networks.

Figure 5 and Figure 6 also show the Gaussian process regression predictions with 25 repetitions. We expect GPR to only

predict the right estimate of the mean of the response. Since it is based on Bayesian inference, which is very different from the back-propagation mechanism used in MDNs, it can infer the response estimate with a much smaller training dataset. The GPR model is not trained on datasets larger than 2500 samples because it scales poorly and the training expense grows exponentially as mentioned in Section 2.1.

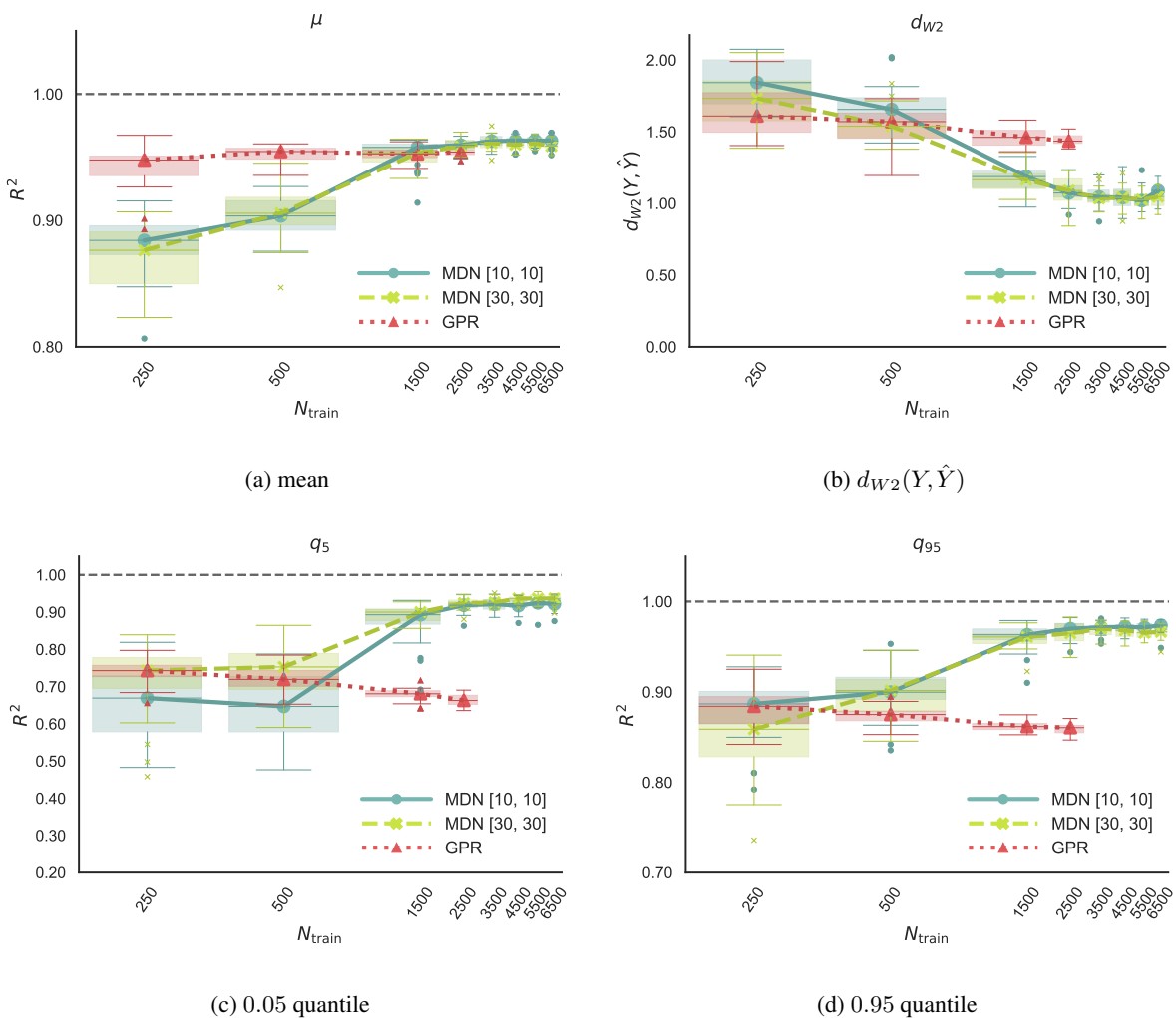

**Figure 6.** CASE-OFFSHORE: Figures showing the change in the normalized 2-Wasserstein distance, $R^2$ value of the mean, 0.05 quantile and 0.95 quantile of the predicted pdf as a function of the training samples. The study is performed on the tower base fore-aft moment standard deviation (TwrBsMyt stddev). The numbers in the square bracket $[x, x]$ denote the widths of the first and second layer of the 2-layer neural networks.

In both CASE-ONSHORE and CASE-OFFSHORE, the difference between the predictions of the two MDN architectures [10, 10] and [30, 30] is negligible for $\mu$, $q_{5\%}$ and $q_{95\%}$. An improvement of roughly $15\%$ in terms of $d_{W2}$ is seen in CASE-ONSHORE, whereas a negligible difference is observed in CASE-OFFSHORE. As we do not have an infinite pool of data, the uncertainty bounds concerning the choice of the training samples progressively reduce as we approach the total available training samples. At smaller $N_{train}$ values, the uncertainty is also driven by the missing information in the training data and the choice of the initial conditions used by the stochastic gradient descent optimizer. Significantly better GPR estimates of the

response mean for less than $1500$ training samples can be attributed to the Bayesian formulation. Beyond that point, however, MDNs and GPR are comparable, with $R^2 > 0.99$ in CASE-ONSHORE and $R^2 > 0.95$ in CASE-OFFSHORE. MDN shows significantly better predictions of all other quantities.

The estimates of the tails of the pdf, quantified by the lower $5\%$ quantile, are well captured by MDN in both onshore and offshore case studies. Overall, the model's accuracy in terms of the statistical quantities is approximately $5\%$ better in CASE-ONSHORE for the same number of training points. However, $d_{W2}$ is $50\%$ smaller in CASE-ONSHORE than in CASE-OFFSHORE, signifying, overall, a much better inference of the latent pdf in the onshore conditions than offshore. This is expected, as larger the degrees of freedom, the larger the space between the neighboring points. The goodness of fit of the model in terms of $d_{W2}$ depends on the rate of change of the statistics of the underlying distributions. The gradient of the standard deviation, in particular, is much harder to infer from a cloud of points than the gradient of the mean. The convergence plots for CASE-OFFSHORE could indicate either that this is the best estimate of $d_{W2}$ achievable, given the information the model has, or that the number of samples needed to capture the variation in the standard deviation are much larger than the maximum training sample size that we tested.

Figure 7 shows the training and validation losses plotted against the number of epochs for CASE-ONSHORE. MDN overfits the data at $N_{train} = 500$ because, as the training loss decreases, the validation loss increases, indicating that the model cannot handle previously unseen data. Figure 7b is well-fitted as the training and validation losses decrease at the same rate. The plot also shows the auto-stop algorithm at work, which halts training after 100 epochs of approximately zero-gradient loss to avoid overfitting.

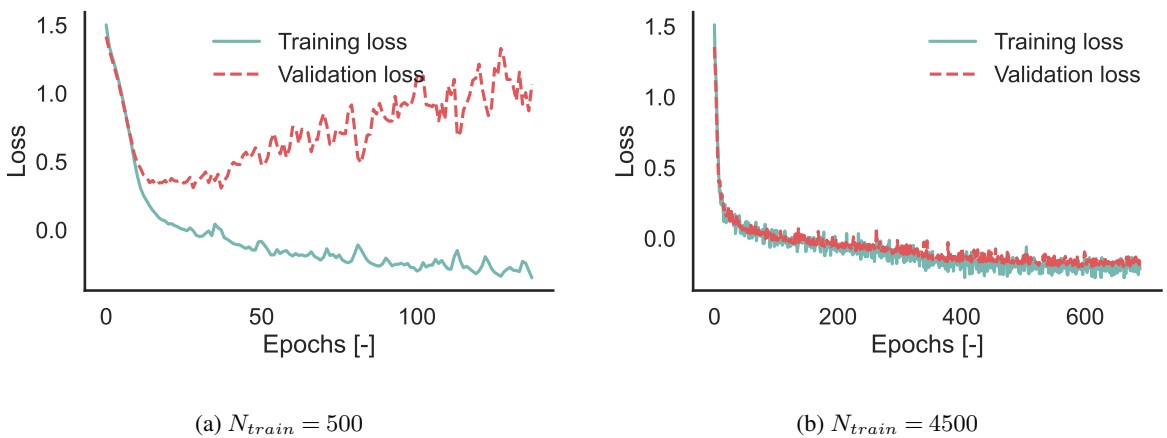

(a) $N_{train} = 500$            (b) $N_{train} = 4500$

**Figure 7.** Training and validation losses for the [10, 10] MDN with 500 (a) and 4500 (b) training samples, CASE-ONSHORE.

For the remainder of this study, we will use a two-layer MDN with ten activation units in each layer trained on $4500$ samples. It offers a good balance between training time, model complexity, and accuracy. For GPR, a training set of $500$ samples will be used as it is found to be sufficiently accurate.

## 4.2 Load prediction

In this section, we evaluate the prediction of the 10-minute damage equivalent loads on the wind turbine for CASE-ONSHORE and CASE-OFFSHORE. Table 5 summarizes the predictions of various statistical properties of the response pdf for both cases in terms of the $R^2$ values. $R^2$ values reflect the proportion of variance in the output that is predictable from the surrogate models. The absolute magnitude of $R^2$ is sensitive to the optimization initialization, choice of the test samples, and the choice of the training subset as seen in Figure 6. It is, therefore, important to note that the absolute $R^2$ values do not carry much objective meaning on their own. They are only used here for comparing the performance of models relative to one another. Figure 8 shows the corresponding plots for CASE-OFFSHORE.

**Table 5.** Comparison of the prediction of the statistical properties of the response pdf for the tower base fore-aft loads. The table lists the $R^2$ values observed for the statistics of the tower bottom fore-aft load channel for the onshore and offshore test cases.

<table>
<tr><td colspan="9" align="center">Tower base fore-aft $DEL^{ST}$ $R^2$</td></tr>
<tr><td></td><td colspan="4" align="center">CASE-ONSHORE $R^2$</td><td colspan="4" align="center">CASE-OFFSHORE $R^2$</td></tr>
<tr><td></td><td>$\mu$</td><td>$\sigma$</td><td>$q_5$</td><td>$q_{95}$</td><td>$\mu$</td><td>$\sigma$</td><td>$q_5$</td><td>$q_{95}$</td></tr>
<tr><td>MDN</td><td>0.994</td><td>0.942</td><td>0.962</td><td>0.991</td><td>0.955</td><td>0.782</td><td>0.879</td><td>0.977</td></tr>
<tr><td>GPR</td><td>0.943</td><td>-0.85</td><td>0.488</td><td>0.819</td><td>0.936</td><td>-0.322</td><td>0.693</td><td>0.852</td></tr>
</table>

Overall, MDN performs better than GPR across all the metrics listed in Table 5 for CASE-ONSHORE and CASE-OFFSHORE. The conditional average, $\mu$, is well estimated by both models. The standard deviation, $\sigma$, is constant in the case of the GPR model, as it is a homoscedastic formulation. This, as expected, is reflected through the negative values of $R^2$, indicating no correlation between the $\sigma$ estimated from the GPR model and the reference $\sigma$ values. The minor variations in Figure 8b in $\sigma_{surrogate}$ can be attributed to a combination of model uncertainty and Monte Carlo sampling. Estimates of the standard deviation by MDN are excellent in CASE-ONSHORE, but the performance drops in CASE-OFFSHORE. However, the results are encouraging compared to GPR and show that MDN can handle heteroscedastic datasets. The 5% and 95% quantiles, which are essential for design considerations, are exceptionally well predicted by MDN. In Figure 8c and Figure 8d, GPR shows a bias in the quantile estimate, increasing with the quantity's magnitude, which can be directly ascribed to the underestimation of the standard deviation of the response.

The $R^2$ values show that the model fit for CASE-ONSHORE is relatively better than for CASE-OFFSHORE. As noted in the previous section, it appears to be much easier to train a case with only aerodynamic features for the same number of training samples and network architecture.

Table 6 compares the $R^2$ values calculated for $DEL^{ST}$ of various load channels on CASE-OFFSHORE by MDN and GPR. Both MDN and GPR show relatively high accuracy in estimating the mean $DEL^{ST}$ for the tower channels. Specifically, the tower bottom and top fore-aft channels have $R^2$ values above 0.94 for MDN, indicating strong predictive performance. However, a notable discrepancy is observed in the GPR predictions for the mean blade root edgewise $DEL^{ST}$. We noted that

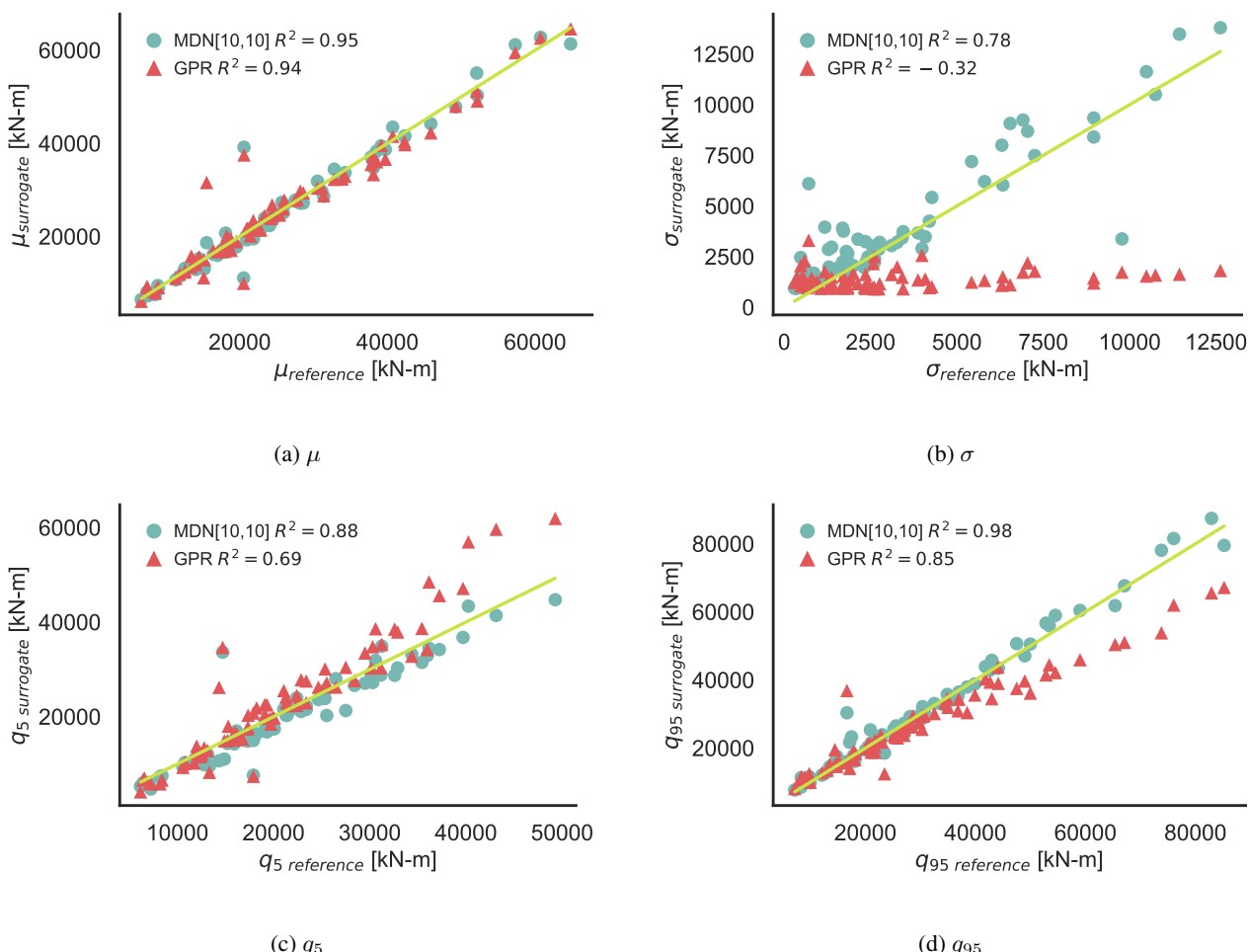

**Figure 8.** CASE-OFFSHORE: Prediction of the statistics of the response pdf of the 10-minute damage equivalent loads for the tower base fore-aft moment.

the GPR predictions were especially sensitive to the value of the inital length scale vector used for training this channel. Further investigations show that up to $18\mathrm{m/s}$, the GPR is able to correctly estimate the mean blade root edgewise $DEL^{ST}$. However, between $18\mathrm{m/s}$ and $20\mathrm{m/s}$, the values are overestimated and above $20\mathrm{m/s}$, they are underpredicted, leading to a smaller $R^2$. This behavior could be attributed to large gradients in the response surface and insufficient data samples at high wind speeds. The conditional response at high wind speed in Figure 12c also shows the potential for this bias due to the presence of a few data samples with very low fatigue values at high wind speed and high turbulence intensity, as the wind turbine switches to the idling regime in some cases.

**Table 6.** CASE-OFFSHORE: Comparison of the channel-wise prediction of the $R^2$ values of the mean and standard deviation using MDN and GPR.

| | MDN $R^2$ | | GPR $R^2$ | |
|---|---|---|---|---|
| | $\mu$ | $\sigma$ | $\mu$ | $\sigma$ |
| Tower base fore-aft $DEL^{ST}$ | 0.95 | 0.78 | 0.93 | -0.32 |
| Tower top fore-aft $DEL^{ST}$ | 0.94 | 0.95 | 0.89 | -0.14 |
| Blade root edgewise $DEL^{ST}$ | 0.915 | 0.85 | 0.48 | 0.03 |
| Blade root flapwise $DEL^{ST}$ | 0.87 | 0.79 | 0.86 | -0.74 |

The $DEL^{ST}$ for the tower base fore-aft moment (Figure 9), tower top fore-aft moment (Figure 10), blade root flapwise moment (Figure 11) and blade root edgewise moment (Figure 12) are plotted at three operational conditions falling in low, medium and high wind speed blocks. The values of the input features are noted in Table 7.

**Table 7.** Values of the input features for the pdfs in Figures 9 to 12.

| Wind condition | $u$ [ms$^{-1}$] | $ti$ [-] | $\alpha$ [-] | $H_s$ [m] | $T_p$ [s] | $wdir$ [deg] |
|---|---|---|---|---|---|---|
| Low wind speed | 6 | 40 | 0.08 | 1.0 | 7.0 | 0 |
| Medium wind speed | 12 | 16 | 0.08 | 1.0 | 7.0 | 0 |
| High wind speed | 21.2 | 18.5 | 0.42 | 2.2 | 11.5 | 148.9 |

Clearly, the responses are not always normally distributed. The variance of the response is not constant across wind speeds. Near the cut-out wind speed, we also notice a multi-modal response, as the wind turbine switches between idling and power production, depending on the local variations in the inflow wind patterns. Here, MDN is shown to leverage the flexibility of learning complex noise patterns to then approximate the full picture of the response that deterministic models could otherwise miss.

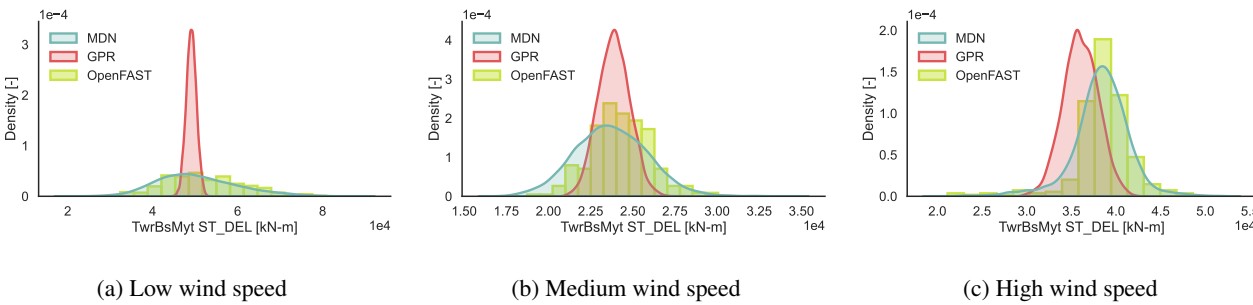

(a) Low wind speed      (b) Medium wind speed      (c) High wind speed

**Figure 9.** CASE-OFFSHORE: Predicted and reference (OpenFAST) conditional pdf for the tower base fore-aft moment $DEL^{ST}$.

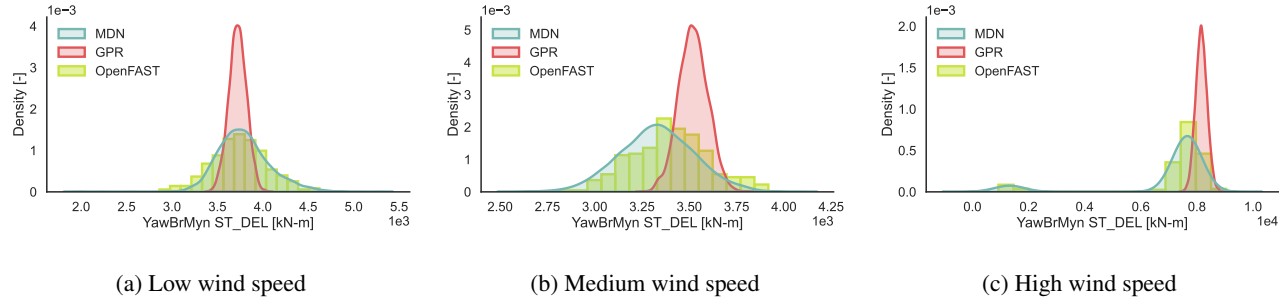

(a) Low wind speed        (b) Medium wind speed        (c) High wind speed

**Figure 10.** CASE-OFFSHORE: Predicted and reference (OpenFAST) conditional pdf for the tower top fore-aft moment $DEL^{ST}$.

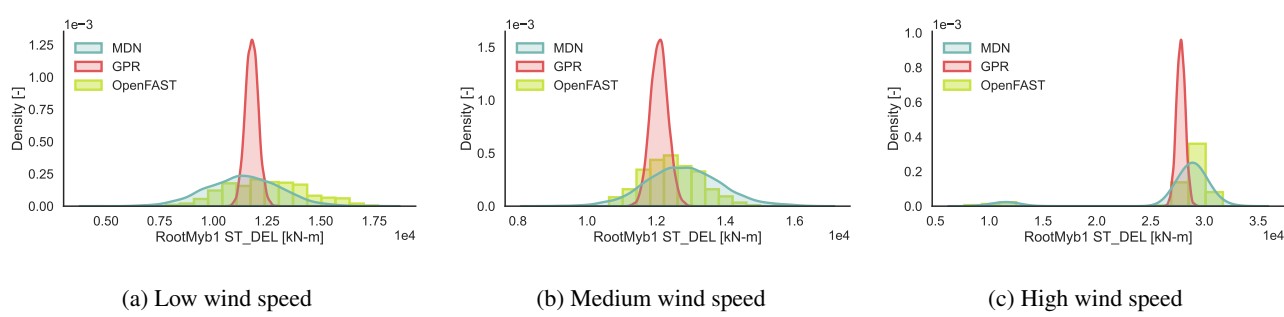

(a) Low wind speed        (b) Medium wind speed        (c) High wind speed

**Figure 11.** CASE-OFFSHORE: Predicted and reference (OpenFAST) conditional pdf for the blade root flapwise moment $DEL^{ST}$.

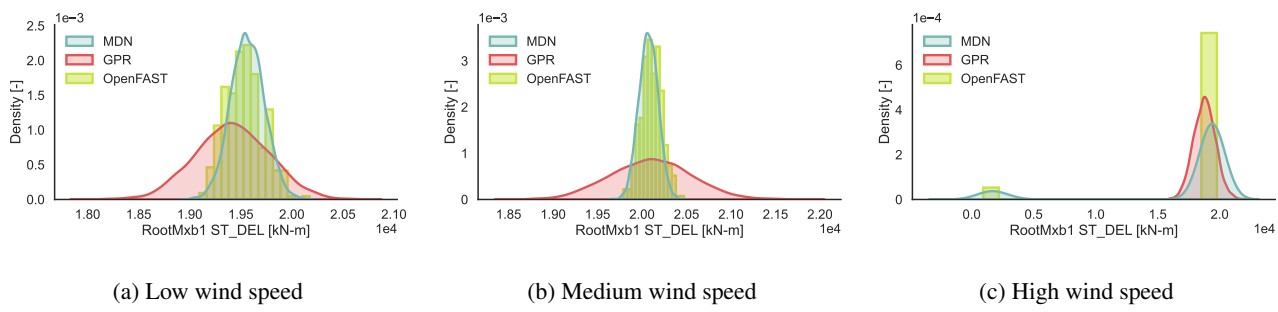

(a) Low wind speed        (b) Medium wind speed        (c) High wind speed

**Figure 12.** CASE-OFFSHORE: Predicted and reference (OpenFAST) conditional pdf for the blade root edgewise moment $DEL^{ST}$.

In Figure 13, $d_{W2}$ is plotted on TEST2 dataset. We notice that there are certain test samples where neither MDN nor GPR is successful in inferring the underlying function. From the figure, it appears that the model is consistently unable to detect the correct patterns at very low turbulence intensities across different wind speeds and load channels. Figure 14 shows the conditional pdf plots for the tower base fore-aft $DEL^{ST}$ at $4\%$ turbulence intensity, corresponding to the high $d_{W2}$ regions in the heatmap in Figure 13a. The figures correspond to wind speeds of 12, 15, and 18 ms$^{-1}$. The other input parameters are

fixed at the values indicated in Table 4. Both GPR and MDN overpredict the standard deviation of the response in this regime, resulting in the relatively high $d_{W2}$ values. Since we do not see a similar decline in performance at the tower top, fatigue driven mainly by hydrodynamic load variation at low turbulence intensities may be difficult for the model to capture. Increased training sample density in this region could benefit performance, but this has yet to be investigated. Application-wise, these
regions are not the most critical, as fatigue is primarily driven by larger turbulent disturbances.

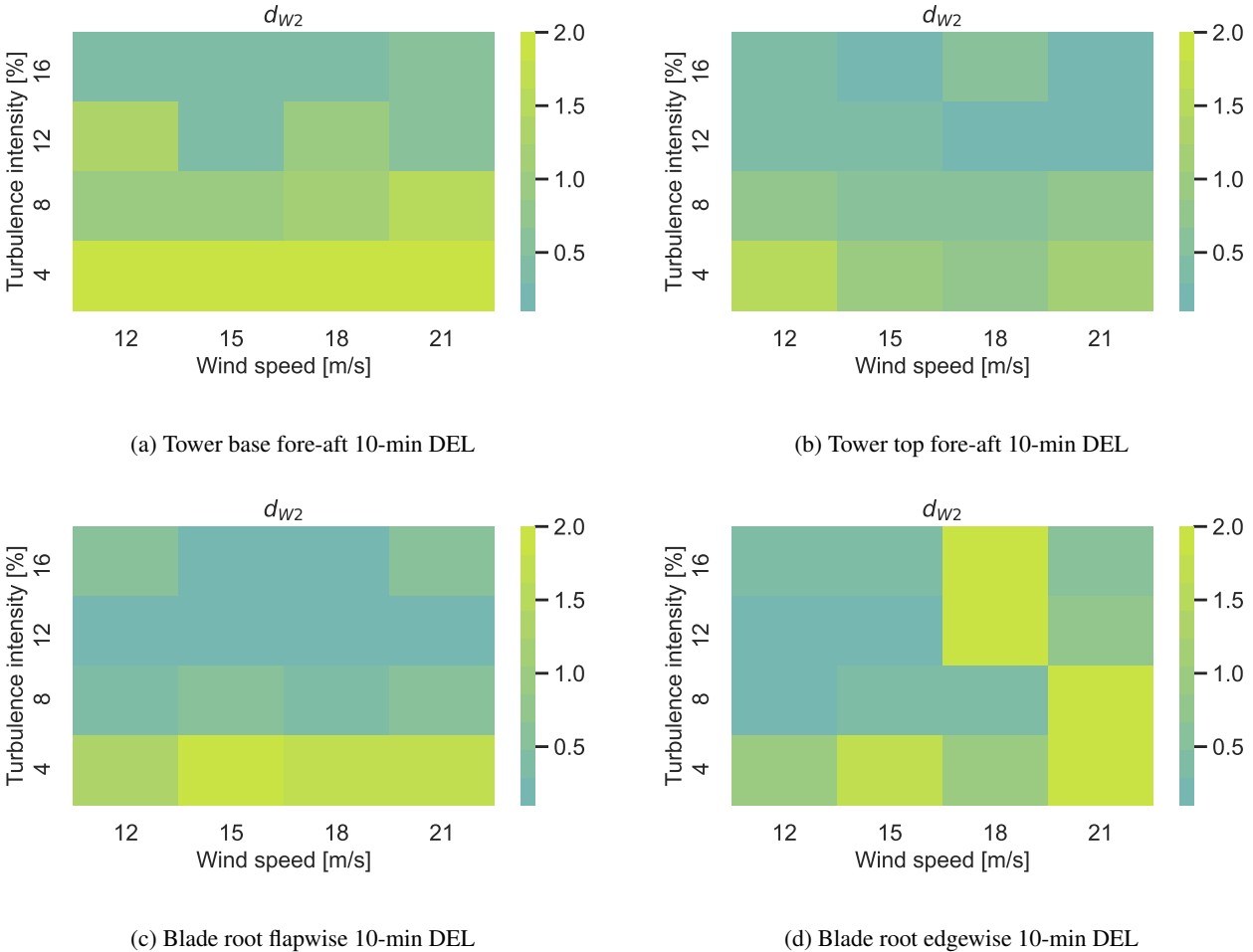

(a) Tower base fore-aft 10-min DEL            (b) Tower top fore-aft 10-min DEL

(c) Blade root flapwise 10-min DEL            (d) Blade root edgewise 10-min DEL

**Figure 13.** Normalized 2-Wasserstein distance computed on CASE-OFFSHORE validation dataset for the MDN model. The performance is plotted on a turbulence intensity and wind speed grid.

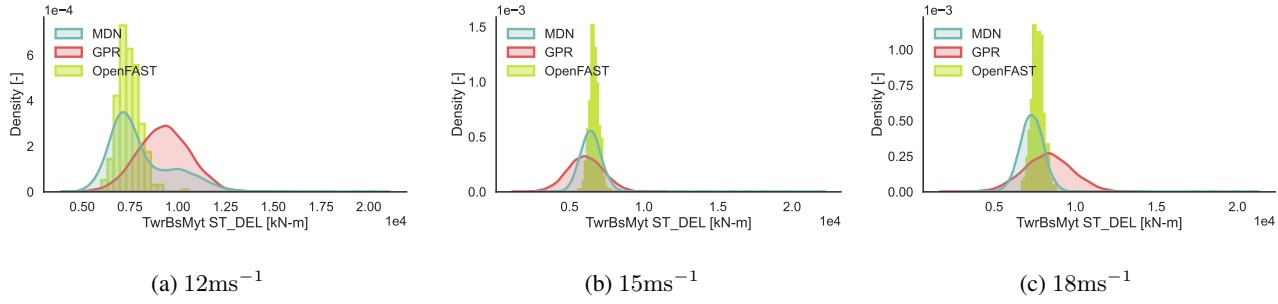

$$\text{(a) } 12\text{ms}^{-1} \qquad\qquad \text{(b) } 15\text{ms}^{-1} \qquad\qquad \text{(c) } 18\text{ms}^{-1}$$

**Figure 14.** CASE-OFFSHORE: Predicted and reference (OpenFAST) conditional pdf for the tower base fore-aft $DEL^{ST}$ at low turbulence intensity (4%) values at wind speed of 12, 15 and $18\text{ms}^{-1}$.

## 5 Conclusions

This paper presents a novel probabilistic approach based on mixture density networks to make efficient and flexible load surrogates for offshore siting. The data-driven surrogate uses aero-servo-hydro-elastic OpenFAST simulations of the 10-MW reference wind turbine for training. We compare the performance of MDN to the widely used Gaussian process regression model and show an improvement in the estimation of the load uncertainty associated with the stochastic representation of inflow turbulence and waves.

The surrogate is trained on a wind turbine subject to aerodynamic (CASE-ONSHORE) and aero-hydrodynamic (CASE-OFFSHORE) loading with the intent of comparing the difficulty in designing load surrogates for the two cases. The reference conditional pdfs for validating the models' performance are produced using 300 random seeds at each of the test operating conditions. A convergence study is performed to assess the accuracy of the surrogate as a function of the number of training samples. Two different MDN architectures and the standard Gaussian process regression are evaluated. It is shown that the surrogate is more accurate for the same number of training samples in CASE-ONSHORE (three features) as opposed to CASE-OFFSHORE (six features), based on the 2-Wasserstein distance between the predicted and the reference conditional pdf of the response. A minimum of 2500 samples are required by MDN to surpass an $R^2$ value of 0.95 for the prediction of the mean and quantiles in CASE-OFFSHORE. The GPR model is shown to be more accurate in predicting the mean of the response even with a small dataset of 250 samples. In applications where the surrogate must be trained frequently on new cases, say, for new geometries of the blade or tower, GPR provides a big advantage in limiting the computational cost of training. Beyond 1500 samples, MDN predictions are consistently better. The quantiles are well captured by MDN in both cases.

The conditional pdfs from the validation dataset are evaluated for low, medium, and high wind speed cases to demonstrate the ability of MDN to capture heteroscedastic, multi-modal responses with high accuracy, even with limited training data. We note a poor performance of the MDN model at low turbulence intensity conditions across all load channels, indicating either the need for a higher sampling rate in those regions in the training dataset or the presence of a sharp gradient in the response surface that the model could not appropriately capture.

This work shows promising results for using MDN and GPR as surrogates in site assessment of onshore and offshore wind
turbines. The MDN model probabilistic modeling of the loads, although shown to have a slight improvement in the prediction
of the expectation of the response compared to the state-of-the-art Gaussian process regression, can capture the variances
and quantiles of the response far better. With the added benefit of not needing seed repetitions prior to training, we show
that both approaches cut down significantly on the computational cost associated with generating the training database without
compromising on the accuracy of prediction. Work is currently in progress to determine how the information on the uncertainty
in the short-term load response can be propagated to the lifetime loads to help inform engineering decisions.

*Data availability.* Datasets related to this article, described in Section 3, can be found at https://doi.org/10.4121/21939995.v1, hosted at
4TU.ResearchData (Singh, 2023).

## Appendix A:  Machine learning framework

Figure A1 shows the basic framework used for model calibration and load estimation. All data generated from OpenFAST is
included in the training base without any repetition or pre-filtering step involved.

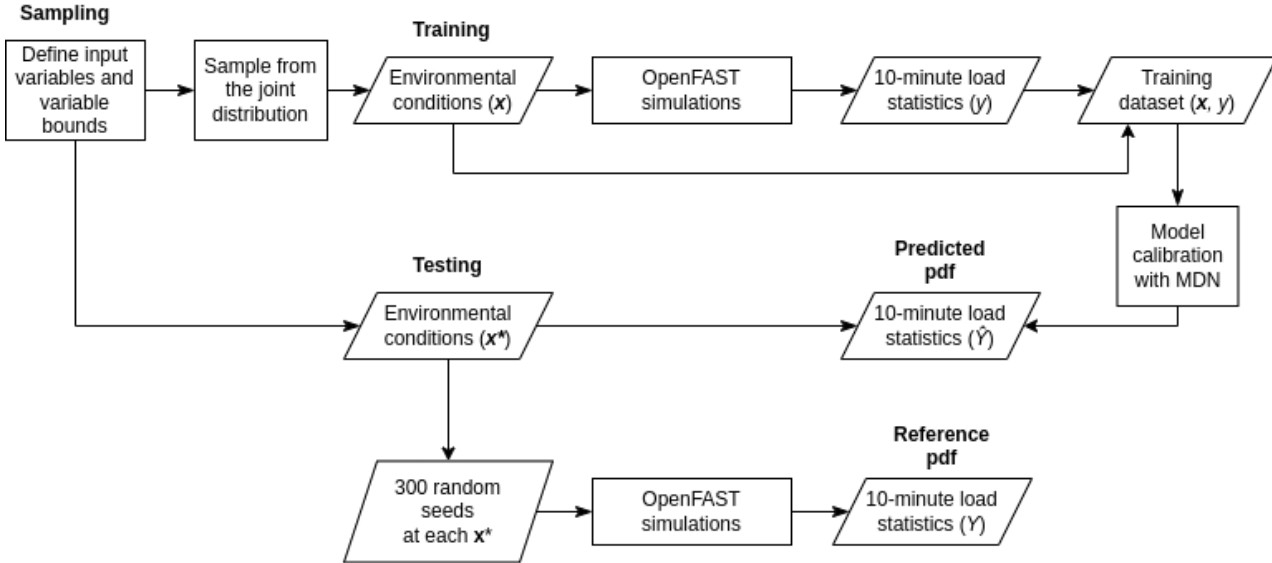

**Figure A1.** Schematic of the machine learning framework.

*Author contributions.* **Deepali Singh:** Conceptualization, Methodology, Software, Validation, Data curation, Investigation, Writing - original draft, Visualization. **Richard P. Dwight:** Conceptualization, Supervision, Methodology, Writing - review & editing, Project administration. **Axelle Viré:** Supervision, Writing - review & editing, Project administration, Funding acquisition.

*Competing interests.* The authors declare that they have no known competing financial interests or personal relationships that could have
appeared to influence the work reported in this paper.

*Acknowledgements.* The project has received funding from the European Union's Horizon 2020 research and innovation programme under grant agreement No. 860737 (STEP4WIND project, step4wind.eu). The authors are grateful to Kasper Laugesen and Erik Haugen (Siemens Gamesa Renewable Energy, Denmark), for their valuable feedback.

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
