# Peer review of "Probabilistic surrogate modeling of damage equivalent loads on onshore and offshore wind turbines using mixture density networks"

_Wind Energy Science, 2024_

## Referee Comment (RC3)

May 31, 2024

**Review of the article:**
**Probabilistic surrogate modeling of damage equivalent loads on onshore and offshore wind turbines using mixture density networks**

**General comments**:

The article addresses the problem of probabilistic surrogates for aeroelastic wind turbine models, a relevant scientific question within the scope of WES and with significant impact on the wind energy industry. The article includes interesting new ideas and techniques in the use of mixture density networks to predict the local distribution of the response of fatigue loads given a set of turbulent wind inflow and sea state parameters.

The article is well structured and written concisely. The scientific methods, analyses and assumptions are valid and clearly outlined and reproducible. The presented results (convergence study with number of aeroelastic simulations, with two out-of-sample test datasets) are sufficient to support the interpretation, discussion and conclusions.

The authors cite sufficient and relevant work in their literature synthesis.

Some points to improve:

- The objectives of the study or hypotheses should be stated in the introduction.
- The original contribution of the paper to the field should be clearly stated in the introduction.

**Specific comments**:

- **L20**: You are missing a general description and references to the most used aeroelastic models: OpenFAST, HAWC2, Bladed, Flex, etc.
- **L32**: The reference to "fast frequency-domain reduced-order models" seems disconnected from the article. It should be either removed, or explained why these techniques are interesting. If these techniques are interesting, why not testing them in the article? Are these ROMs a way to replace the aeroelastic model (OpenFast)? Are the probabilistic surrogates presented in the article relevant for ROMs. I do not necessarily agree to describe ROMs as physics-based models, as they can be trained on simulations just as your load surrogates. Are aeroelastic models not physics-based?
- **L42**: The uncertainty due to the presence of unknown or inexpressible features is an epistemic uncertainty. Epistemic uncertainties represent lack of knowledge and therefore they could be reduced by for example: improving the accuracy of a measurement, or increasing the number of degrees of freedom (DOF) used to characterize a phenomena. If you represented the uncertainty in the turbulent-wind and sea-state field with more degrees of freedom (let's say a grid with three wind/wave components) you would not have this uncertainty. Since you use few integrated quantities to represent the inflow you have the epistemic uncertainty of the flow realization.
- **L57**: TurbSim is not a turbulence model, but a software. You should refer to the turbulence model used, i.e. Mann? KAimal spectra? etc. You should also report the parameters used on the spectra.
- **L68**: Deterministic models can be used to build probabilistic surrogates similar to the ones you propose: Quantile regression proposes the use of multiple deterministic surrogates to

predict the local distribution (Koenker, Roger, and Kevin F. Hallock. 2001. "Quantile Regression." Journal of Economic Perspectives, 15 (4): 143-156. DOI: 10.1257/jep.15.4.143) (Meinshausen, N. and Ridgeway, G., 2006. Quantile regression forests. Journal of machine learning research, 7(6).). You should add a sentence here on this. Note that these types of probabilistic surrogates need to have multiple seeds in the training in order to compute the local quantiles.

- **Figure 1**: The deterministic model should also have the explicit dependency on the inputs $\hat{y}(\mathbf{x})$.

- **L96**: This sentence needs to be re-written for more clarity. These probabilistic models can be train without having to simulate multiple inflows for each input vector, because the methods can infer the local distribution based on the samples available in the neighborhood. This point is later clarified in L240-L242. But is should be clear from the beginning.

- **L265**: Missing citation to Sobol.

- **L270**: The distribution of wave state parameters is fitted using KDE based on what? The following text and Table 2 are confusing because you focus on the lower and upper bounds, but this is not relevant for the KDE.

- **Table 2**: The distribution of the turbulent wind field parameters is reported to be uniform in table 2, but seems not to be the case on the marginal distributions on Figure 3. This table should be accurate description of the simulations.

- **Table 2**: Why use a random uniform distribution of seeds and not just an increasing seed number? Consecutive seed numbers will provide a similarly independent field as "random" seeds.

- **Table 3**: The symbol $\Delta$ needs to be defined as the discretization step.

- **Figure 3**: The marginal distributions should be presented as histograms and not as KDEs as it gives a wrong sense of the actual distribution.

- **Figure 3**: The marginal for the shear exponent should be re-scaled to show the distribution of the train and test 1, instead of focusing on test 2.

- **Figure 4**: The outliers at high wind speed that give low moments (and that cause the bi-modal local distributions) should be highlighted and explained. Why do they occur?

- **L336**: It is not clear why you are using the standard deviation instead of the damage equivalent load for the convergence study. I expect the DEL to be harder because of the power exponent. This should be corrected.

- **Table 4**: The label should explicitly state that these are $R^2$ results.

- **L407**: The description of the reference local distribution should be given in the methodology, i.e. table 3. and not here.

- **L418**: The conclusion of the causes of the low performance at low TI should be illustrated by showing the local distributions, as you did in figures 11 or 12.

---

## Author Response (AR1)

**Authors' response**

Dear Editor, Reviewers,

We deeply appreciate the opportunity to submit a revised version of the manuscript titled "Probabilistic surrogate modeling of damage equivalent loads on onshore and offshore wind turbines using mixture density networks" to Wind Energy Science.
5 We thank the reviewers for their constructive, detailed and insightful comments on our manuscript and the time and effort they have dedicated to providing useful feedback. We are certain that their input has contributed greatly to the quality and clarity of the article.

The reviewer's comments have been listed in order below in italic text, followed by our responses in normal text. The modifications are highlighted in the manuscript in teal. Please find a point-by-point response to the reviewers' comments and
10 concerns below.

Best regards,
Deepali Singh, Richard P. Dwight, Axelle Viré

**Reviewer 1**

*Q 1.1 In Section 3.3, it is mentioned that "The 10-minute fatigue is calculated using short-term damage equivalent loads*
15 *($DEL^{ST}$)". But, Section 4.1 says "The network is trained on the tower base fore-aft moment standard deviation (TwrBsMyt [kN-m] stddev)". It is not clear what are the inputs and outputs of the surrogate. Please formulate the surrogate model formally and mathematically.*

**Reply**:

- The reviewer is right in noting that the output list is not mentioned clearly apart from a brief mention in Section 3.3.
20    We have included an additional explanation in Section 3.3, page 14, line 641, to make the output list more explicit. The model is trained on the 10-min. statistics including the standard deviation and $DEL^{ST}$.

- In Table 2 a new column has been included to indicate which input variables are used for each case.

- An additional explanation has been added in Section 4.1 to clarify why TwrBsMyt [kN-m] stddev was chosen for the convergence study. Essentially, we noted that this channel was particularly difficult to train, with the highest average
25    $d_{W2}$ value among all channels. Models are data-dependent, and each channel ideally requires a separate sensitivity study on the number of training samples needed for convergence, the model architecture, and the model hyperparameters. However, for practical purposes, we assume that if a model can perform well on the channel with the poorest metrics, it should also perform well, if not optimally, on the other load channels using the same network architecture.

*Q 1.2 Section 3.5 Accuracy metric should be better explained with more detailed information. Equation 18: It is not clear if*
30 *the summation is over the whole test dataset or over the seed repetitions of each test sample. How many samples are drawn from the MDN output distribution to compute the statistics (mean, std, and the quantiles)? Equation 19: How are the quantile functions computed, by combining closed-form formulations for each Gaussian or through sampling directly from the output mixed distribution?*

**Reply**: Text with additional explanation about the Wasserstein distance and its calculation has been added to page 17, line 687.
35 The text covers answers to the three questions posed by the reviewer. In summary,

- The summation is over the entire test dataset. It is reflected in Equation (20).

- We used 5000 samples from the MDN output to calculate the distribution statistics.

- The quantile functions for calculating the Wasserstein distance are obtained from the samples of the empirical distributions: 5000 samples from MDN output and 300 from the seed repetitions in OpenFAST.

**Q 1.3** *The seed repetitions with the same input variables are not necessary since the prediction is inferred from neighboring inputs, however, it also means that many neighborhood points are still required to have a good estimate of aleatory uncertainty. In addition, when modelling in high dimensional input space, the actual distance between the test point and the input points might be far. What is the robustness of the model in such case?*

**Reply**: As the reviewer correctly noted, the larger the degrees of freedom, the larger the space between the neighboring points. The goodness of fit of the model would depend on the rate of change of the statistics of the underlying distributions. The gradient of the standard deviation, in particular, is much harder to infer from a cloud of points than the gradient of the mean. A small dataset could result in biased estimates of the standard deviation.

In practice, we see in this case that the prediction accuracy is higher in the case of the onshore turbine with only three features as opposed to the offshore case with six. The convergence plots Figure 7b and Figure 8b show the models approaching convergence at different rates for the two cases but appear to have robust $d_{W2}(Y, \hat{Y})$ estimates once the training sample size is above 3500. It can also be seen that $d_{W2}(Y, \hat{Y})$ converges to a higher value in the offshore case, and it is not evident that adding more training samples or using a larger network increases the accuracy of prediction. The convergence, in the offshore case, could indicate either that this is the best estimate of standard deviation achievable, given the information the model has, or that the number of samples needed to capture the variation in the standard deviation is much larger than the maximum training sample size that we tested. This could be an interesting future follow-up of this study.

On page 22, line 753, this explanation has been included in the text.

**Q 1.4** *Line 215 "It pushes the coefficients of uninformative features towards zero, effectively pruning the feature space." Additional explanations or references would improve the reader's comprehension.*

**Reply**: A reference has been added on page 10, line 572 to the LASSO paper which contains more information about the regularization methodology. We also amended the section to replace lower case characters $l1$ and $l2$ to $\lambda$ for $L1$ and $L2$ as it was unnecessary to introduce new symbols here.

**Q 1.5** *Table 1. What is the sensitivity of mixture component numbers in terms of accuracy and computational resources?*

**Reply**: We performed a sensitivity study to the number of components with 2, 4, 12, and 20 components for the tower bottom fore-aft $DEL^{ST}$ channel with 4500 training samples. Figure 1 shows the normalized 2-Wasserstein distance values plotted as a function of the number of mixture components the MDN was trained on. The MDN consisted of two layers with ten neurons each. The box plots are based on 10-fold training to quantify the uncertainty associated with the choice of the training subsample. The difference in the predictions is not very significant and seems to be robust against changes in this hyperparameter.

**Q 1.6** *Please describe what is the specific purpose of having two different test datasets TEST1 and TEST2.*

**Reply**: TEST1 is a randomly sampled subset of the training space, whereas TEST2 is an ordered sample. The intention with TEST2 was to be able to plot heatmaps for turbulence intensity vs wind speed, such as the plots in Figure 15. This explanation is included on page 15, line 664.

**Q 1.7** *Figure 3: Why is the marginal distribution of alpha a deterministic value only? Isn't it a uniform distribution for the 'train' and 'test 1' according to Table 2?*

[Figure]

**Figure 1.** The normalized Wasserstein distance for the tower bottom fore-aft $DEL^{ST}$ trained on 4500 training samples, as a function of the number of mixture components.

**Reply**: The reviewer is right to note this, as it is an artifact of the pairplot due to the samples in TEST2 taken at a single value of $\alpha$. The pairplot in Figure 4 is replaced to show only the marginal distribution histograms of the training data set in Figure 5.

***Q 1.8*** *Figure 6: Please define the spread of the boxes and whiskers (within which quantiles?).*

**Reply**: The boxes and whiskers follow the Matplotlib standards. For clarity, the definitions are now explicitly addressed on page 19, line 718. The boxes extend between the first ($Q1$) and third ($Q3$) quartile of the data, and the horizontal line across the box indicates the median. The difference between $Q1$ and $Q3$ defines the interquartile range ($IQR$). The upper whisker extends to the largest data values that are within $1.5IQR$ above $Q3$. The lower whisker, similarly, extends to the lowest data point within $1.5IQR$ below $Q1$. The outliers are visible as dots beyond the whisker boundaries.

***Q 1.9*** *Line 174 "L-BFGS-B algorithm" Limited memory Broyden–Fletcher–Goldfarb–Shanno algorithm for simple bound constraints is not defined.*

**Reply**: The full form has been added to the text on page 8, line 530, as requested.

**Reviewer 2**

***Q 2.1*** *Clarify the difference between design standards and site specific analysis*

**Reply**: We have modified the introduction to clarify the difference between certification and site-suitability assessment on page 2, line 339. The site suitability assessment provides the assurance that the environmental conditions that affect the loading, durability and operation of the turbines are in accordance with the design. This is a preliminary analysis performed before designing a site-specific tower. Certification based on design standards involves a more rigorous analysis, for which only industry-standard engineering tools are used. The surrogate is not intended to perform certification assessment, but rather provide a quick estimate of the loads for any site, using the design standards as a framework to define the DLCs.
Section 1 on page 2, line 339 has been modified to clarify the distinction to the reader.

***Q 2.2*** *What is being "designed" during site analysis? Towers and foundations? Or just checking stock design of rotor/nacelle. Clarify for the reader a bit more at the start. Also has implications; eg if we find we need to change the tower design, have to rebuild the surrogate (which is expensive!)*

**Reply**: In this study, we are not designing any part using the surrogate. However, the framework could be extended to include geometric parameters, soil properties, water depth, and other factors, depending on the application. We are using the surrogate solely to estimate loads on a wind turbine where the nacelle and blades are already fixed in terms of design, with a reference tower.

Linked to the previous comment of the reviewer, the paragraph on page 2, line 339 in Section 1 provides more clarification on the context for which the surrogate is used.

*Q 2.3 Include references/discussion to older freq domain model to show how long been around/lineage*

**Reply**: As a response to reviewer 3, we have decided to remove the line about the frequency domain models from the text. This is because the frequency domain models are not used for estimating fatigue loads and may create confusion for the reader.

*Q 2.4 Are 10 minutes simulations enough for off-shore turbines? Standards specific length to include important wave impacts*

**Reply**: The surrogate may also be trained on simulations that are longer than 10 minutes if the resources allow it. Our aim was to quantify the variation in loads due to uncertainty in inflow conditions, typically addressed in certification requirements with 10-minute simulations using six seeds or a single 60-minute simulation. Long simulations to capture very low-frequency sea states do not significantly impact tower or blade loads in fixed-bottom cases. However, they are crucial for floating wind turbines, particularly for estimating mooring line fatigue. In this paper, we evaluate the uncertainty estimated by the surrogate against 300 random initializations of the inflow parameters, thereby covering a very wide range of variation per test case. Therefore, we believe 10-minute simulations are sufficient for this application.

*Q 2.5 Around line 35 would be nice to have more discussions of the importance of linearity and nonlinear. Contrast what linearized models miss of the physics vs data driven models*

**Reply**: We have included more information on page 2, line 351 about the non-triviality of modeling fatigue loads, which better explains the benefit of using data-driven models for this application.

New text: Modeling the relationship between the $DEL^{ST}$ and environmental conditions is non-trivial. Fatigue is a multiscale phenomenon that depends on the material composition, composite structure, part dimensions, and dynamic inflow. This makes it challenging to model using low-fidelity physics-based approaches. As a consequence, data-driven surrogate models can be beneficial, as they do not require prior knowledge of the underlying physics and can infer complex relationships from observations alone. Especially in systems where analytical closed-form solutions are intractable or the physical properties cannot be easily modeled (Jiang et al., 2020), data-driven surrogates provide a great advantage.

*Q 2.6 Is it just aleatoric uncertainty? It's true the physical system itself has embedded uncertainty, but in the context of a surrogate the choice and extent of the feature selection and training dataset represent epistomalogical sources of uncertainty.*

**Reply**: Aleatoric uncertainty occurs due to unpredictable variation in the features or in the system's performance which cannot be reduced with the addition of more knowledge. On the other hand, epistemic uncertainty can, in principle, be eliminated with sufficient knowledge, hyperparameter tuning and expert judgements. Although we try to identify the sources of uncertainty, it is quite difficult to differentiate between them from the surrogate model's predictions as they are often combined.

The reviewer's comment refers, first, to the epistemic uncertainty associated with the choice of the features used for training. The site conditions are limited to a certain number of parameters and are generally only available as aggregates. In the context of simulation-based training, we control the parameters that change between each simulation. Therefore, the variation seen in the fatigue response in our dataset can only be caused by variations in one of the chosen features. We assume that we have carefully selected the features affecting the fatigue loads in reality, considering what is typically measured at a representative site. A second way of adding more knowledge is to train the model on the same number of features but provide the entire time series of the wind and waves at different grid locations as input. This could reduce the uncertainty in the response (epistemic), but it is not possible to obtain all this information in practice, especially for the application for which the surrogate is designed.

To summarize, because we are limited by the amount of knowledge we have about the site (number of features and time resolution), the uncertainty in the fatigue response is considered primarily aleatoric.

The reviewer also refers to the presence of epistemic uncertainty due to the choice of training dataset. This can be investigated to an extent by performing a convergence study up to a point where there is no significant improvement in the predictions at locations where the model is tested, as is done in Section 4.1. We show that adding more training data samples does not improve the predictions, thus the epistemic uncertainty associated with the number of training samples is not significant, although not zero.

Finally, we cannot quantify the epistemic uncertainty associated with the choice of hyperparameters with the standard neural network framework. However, we have performed hyperparameter sensitivity studies to minimize this uncertainty and ensure the robustness of the model.

The text in Section 1 has been restructured as a response to other comments, on page 3, line 374, and no longer explicitly mentions the uncertainty type.

*Q 2.7 Section 1.1; if we know the pdf of Y, how does that get used for design verification? E.g. what do we to get cumulative lifetime fatigue and extreme ultimate loads extrapolation? Nice to have clarity on model usage there, beyond being able to provide variance and pdf info from the surrogate*

**Reply**: The main purpose of a surrogate of such kind is to reduce the training cost associated with seed repetitions. In addition, it provides estimates of uncertainty with respect to loads and a more robust estimate of mean response. In current design or certification frameworks, there is no requirement for the quantification of the uncertainty of the loads. However, Li and Zhang (2019) have extrapolated the 50-year accumulated fatigue damage in the mooring lines, tower etc. which could be beneficial for cost reduction when designing the site-specific tower. As we move towards a reliability based design process in the future, probabilistic surrogates will prove to be useful.

*Q 2.8 Explain what heteroscedasticity actually is and where is comes from in wind turbine context*

**Reply**: The heterogeneity in the variance of the conditional pdf across different operating conditions is known as heteroscedasticity.

The definition of heteroscedasticity is included now on page 3, line 405. The sentence just before this is modified to make the link between wind turbine load response and heteroscedasticity clear.

*Q 2.9 Line 105 add ref to non-heteroscedastic paper. Also later in section talk about Kriging and GPR; clarify these are essentially the same*

**Reply**: The clarification has been included on page 6, line 457. The non-heteroscedastic Kriging papers in the context of wind turbine load prediction are listed in the following line.

*Q 2.10 Section 2.1; mention GP challenges in terms of passing though (or not) the training points depending on detailed implementation and variants of eqn 2. See also line 344*

**Reply**: Computing the inverse of the dense covariance matrix $K_{XX}$ in GPR calculations is expensive, with a computational complexity of $\mathcal{O}(n^3)$ and a memory complexity of $\mathcal{O}(n^2)$ (Rasmussen and Williams, 2006). Due to these computational limitations, GPR models are typically not scaled to very large training datasets. We had to, therefore, constrain our model to 2500 training samples in Section 4.1. This explanation is now reflected on page 8, line 522.

*Q 2.11 Line 221; l1 and l2 variables aren't actually defined in the eqns above? Meaning lambda values?*

**Reply**: Correct, they are the same as the lambda values and it was unnecessary to introduce new variables in the section. The text is now amended on page 10, line 578 to clarify that we are referring to $\lambda$ of the $L1$ and $L2$ regularization functions.

***Q 2.12*** *Fig 2 implied hidden layers of various widths, but table 1 only defines 2 hidden layers not their widths*

**Reply**: We have updated Table 1 with the network width hyperparameter and added additional text on page 10, line 583 to explain that 2 network architectures were investigated in subsequent sections.

180   ***Q 2.13*** *Below eqn 17 mention m=10 for blades but before said not looking at blade loads?*

**Reply**: We are considering also the blade loads, as shown in figures Figure 14 and Figure 13.

***Q 2.14*** *Line 298; not clear what Test2 is. "alter only wind speed and turb;" but still creating how many new sims?*

**Reply**: We have updated the text on page 15, line 664 to include the number of simulations (24) that comprise TEST2.

***Q 2.15*** *Fig 3 redo axis labels. Eg "speed" is windspeed? I'd suggest using the Greek variables directly to avoid confusion*

185   **Reply**: We have redone plot Figure 5 to update the labels to greek letters that match the notation used in Table 2.

***Q 2.16*** *Fig 5. Say in caption what MDN [ x, x] denote. What is x?*

**Reply**: The caption is updated in Figure 7 and Figure 8 to mention that the numbers in the square brackets indicate the width of the first and second layer of the 2-layer neural network.

***Q 2.17*** *In the results it's not clear how GPR is being used. Eg fig 5; does that 4 separate GPRs trained independently, one for*
190   *each predicted quantity? I can't see where this is explicitly in results section*

**Reply**: The quantities shown in Figure 8 are derived from the conditional pdf predicted by one single GPR. We added information on page 17, line 683 and page 17, line 687 to explain that the quantities used for judging the accuracy of the model are derived empirically from the conditional distribution predicted by the model for each test case.

***Q 2.18*** *Around line 370 confused if the r squared value is so dependent on choice of training subset, what is table 4 actually*
195   *showing us? Is that for an average across multiple training sets?*

**Reply**: Around line 370, what we meant to say was that the absolute value of $R^2$ would certainly be different if we used a different test dataset, or 20 or 200 test samples instead of TEST1+TEST2. It is still, however, an important measure of accuracy if two models were to be compared on the same exact test database. This is the purpose of showing the values in Table 5.

***Q 2.19*** *Should make mention in table 4 for Sigma of the GPR you have negative values. This is a typical for me statistical*
200   *definition of r squared so good to remind the reader what this means in the context of training and test data*

**Reply**: We have updated Section 4.2, page 23, line 777, with information about the negative $R^2$ values that indicate no correlation between the GPR model's $\sigma$ predictions and the reference $\sigma$ values from OpenFAST. This behavior is in line with the expectations, as standard GPR cannot follow the heterogeneity in the variance/standard deviation.

***Q 2.20*** *Would be nice to see a bit more discussion on how the results changed for the different load channels you're looking at*
205   *EG tower bottom versus later bending*

**Reply**: We have included Table 6 with the comparison of $R^2$ values for GPR and MDN for various load channels. On page 24, line 788, we elaborate further on the difference in the results between the load channels.

***Q 2.21*** *Figure 7 and the discussion would indicate some optimum number of samples between 500 and 4500. The GPR uses*
*500. The conclusions talk about reduced simulation needs with the proposed mdn method, so it would be nice to revisit what*
210   *practicalities these numbers of samples and plies in terms of creating a database. Back to the use case in terms of how the*
*model might need to be rebuilt when changing tower heights, etc in intended use cases for the tool.*

**Reply**: This is a good addition to the conclusion and we have modified the text in Section 5, page 28, line 835.

**Reviewer 3**

***Q 3.1*** *The objectives of the study or hypotheses should be stated in the introduction. The original contribution of the paper to*
*the field should be clearly stated in the introduction.*

**Reply**: Section 1 on page 3, line 382 has been updated to state the objectives of the study and the novelty of the methodology.
New text:The primary objective of this study is to develop a probabilistic data-driven surrogate that maps 10-minute statistics of wind and wave conditions ($X \in \mathbb{R}^6$), to the corresponding 10-minute load statistics including $DEL^{ST}$ and standard
deviation ($Y \in \mathbb{R}$). We introduce a novel framework for load surrogate modeling that reduces training costs by eliminating seed
repetitions without sacrificing prediction accuracy. This probabilistic modeling approach allows for uncertainty propagation
and quantification, enabling informed decision-making. In this study, we compare the performance of a highly flexible machine
learning method, the mixture density network (MDN) (Bishop, 1994a), with the widely used Bayesian approach of Gaussian
process regression (GPR). The surrogates are tested using the onshore and offshore versions of the IEA 10-MW reference
wind turbine. The load for training the surrogate are calculated using an open-source, multi-fidelity, multi-physics solver called
OpenFAST (NREL, 2022; Jonkman, 2013).

***Q 3.2*** *L20 You are missing a general description and references to the most used aeroelastic models: OpenFAST, HAWC2,*
*Bladed, Flex, etc*

**Reply**: References to some of the models have been included on page 2, line 340 in Section 1.

***Q 3.3*** *L32: The reference to "fast frequency-domain reduced-order models" seems disconnected from the article. It should be*
*either removed, or explained why these techniques are interesting. If these techniques are interesting, why not testing them in*
*the article? Are these ROMs a way to replace the aeroelastic model (OpenFast)? Are the probabilistic surrogates presented in*
*the article relevant for ROMs. I do not necessarily agree to describe ROMs as physics-based models, as they can be trained on*
*simulations just as your load surrogates. Are aeroelastic models not physics-based?*

**Reply**: We agree that the frequency models seem out of place for load emulation. Initially, the idea was to give general
examples of models that can solve for interesting quantities quicker than time domain aeroelastic models. However, this does
create confusion about the objective of this study. We have removed the section about the frequency domain solvers and
reframed Section 1 for clarity.

***Q 3.4*** *L42: The uncertainty due to the presence of unknown or inexpressible features is an epistemic uncertainty. Epistemic*
*uncertainties represent lack of knowledge and therefore they could be reduced by for example: improving the accuracy of a*
*measurement, or increasing the number of degrees of freedom (DOF) used to characterize a phenomena. If you represented*
*the uncertainty in the turbulent-wind and sea-state field with more degrees of freedom (let's say a grid with three wind/wave*
*components) you would not have this uncertainty. Since you use few integrated quantities to represent the inflow you have the*
*epistemic uncertainty of the flow realization.*

**Reply**: In the context of the application of site analysis, we generally only have access to site data in terms of 10-minute
statistics of wind and wave parameters. If we are restricted in a practical sense to provide only the integrated quantities to the
model, then the intrinsic randomness becomes an irreducible uncertainty- i.e., it does not decrease no matter how many training
samples we provide. Therefore, we referred to it as aleatoric in this paper. Please also refer to the response provided to reviewer
2, comment 6 (Q2.6).

***Q 3.5*** *L57: TurbSim is not a turbulence model, but a software. You should refer to the turbulence model used, i.e. Mann?*
*KAimal spectra? etc. You should also report the parameters used on the spectra.*

**Reply**: We have updated the text on page 3, line 399 and also included more information about the spatial coherence parameters and the boundary layer stability in Section 3.

*Q 3.6* *L68: Deterministic models can be used to build probabilistic surrogates similar to the ones you propose: Quantile regression proposes the use of multiple deterministic surrogates to predict the local distribution (Koenker, Roger, and Kevin F. Hallock. 2001. "Quantile Regression." Journal of Economic Perspectives, 15 (4): 143-156. DOI: 10.1257/jep.15.4.143) (Meinshausen, N. and Ridgeway, G., 2006. Quantile regression forests. Journal of machine learning research, 7(6).). You should add a sentence here on this. Note that these types of probabilistic surrogates need to have multiple seeds in the training in order to compute the local quantiles.*

**Reply**: This is now specified on page 4, line 412. We mention in the text now that deterministic models are used for modeling conditional statistics such as the mean, variance and quantiles of the conditional distribution.

*Q 3.7* *Figure 1: The deterministic model should also have the explicit dependency on the inputs y^(x).*

**Reply**: Figure 2 is modified to be consistent with the notation in the rest of the paper, so we don't explicitly mention the dependence on $x$ in the notation anymore.

*Q 3.8* *L96: This sentence needs to be re-written for more clarity. These probabilistic models can be train without having to simulate multiple inflows for each input vector, because the methods can infer the local distribution based on the samples available in the neighborhood. This point is later clarified in L240-L242. But is should be clear from the beginning.*

**Reply**: On page 5, line 440, we have added the following lines to the text: The probabilistic surrogate modeling approaches can be trained without having to simulate multiple seed repetitions for each operating condition, as they can infer the conditional response based on neighboring samples. The training time can, therefore, be significantly shorter compared to existing frameworks.

*Q 3.9* *L265: Missing citation to Sobol.*

**Reply**: On page 12, line 627, the citation to the original paper where Sobol sequencing was introduced has been added.

*Q 3.10* *L270: The distribution of wave state parameters is fitted using KDE based on what? The following text and Table 2 are confusing because you focus on the lower and upper bounds, but this is not relevant for the KDE.*

**Reply**: The wave state parameters are based on joint scatter from some representative sites. The min and max in Table 2 just indicate the minimum and maximum values of $H_s$ and $T_p$ in the scatter. Agree that this could be confusing for the reader, but perhaps it is useful to know the range of values.

*Q 3.11* *Table 2: The distribution of the turbulent wind field parameters is reported to be uniform in table 2, but seems not to be the case on the marginal distributions on Figure 3. This table should be accurate description of the simulations.*

**Reply**: The samples for the turbulent wind parameters are uniformly sampled jointly in three dimensions. That's why the marginal distribution of the wind speed shows more data samples at low wind speeds, as the bounds for the turbulence intensity and shear exponent are not uniform.

*Q 3.12* *Table 2: Why use a random uniform distribution of seeds and not just an increasing seed number? Consecutive seed numbers will provide a similarly independent field as "random" seeds.*

**Reply**: Agreed. Seed numbers could have been sequential too.

***Q 3.13*** *Table 3: The symbol $\Delta$ needs to be defined as the discretization step.*

**Reply**: $\Delta$ is now defined in the caption of Table 4.

***Q 3.14*** *Figure 3: The marginal distributions should be presented as histograms and not as KDEs as it gives a wrong sense of the actual distribution.*

**Reply**: The figure is updated to Figure 5 and just reflects the marginal distributions of the training data with histograms instead of KDE.

***Q 3.15*** *Figure 3: The marginal for the shear exponent should be re-scaled to show the distribution of the train and test 1, instead of focusing on test 2.*

**Reply**: The figure is now updated to Figure 5 and now just reflects the marginal distributions of the training data.

***Q 3.16*** *Figure 4: The outliers at high wind speed that give low moments (and that cause the bi-modal local distributions) should be highlighted and explained. Why do they occur?*

**Reply**: At high wind speeds in the simulations, we found at some conditions, the regimes jumped between normal operation and shutdown beyond cut off wind speed. This is reflected in the loads that either show normal loading or close to zero, because the wind turbine is no longer operating.

***Q 3.17*** *L336: It is not clear why you are using the standard deviation instead of the damage equivalent load for the convergence study. I expect the DEL to be harder because of the power exponent. This should be corrected.*

**Reply**: The performance was very similar for DEL and standard deviation, with slightly less accurate predictions for standard deviation- perhaps due to a larger scatter. This was the reason for choosing standard deviation. We have added clarification text in Section 4 to explain why standard deviation was used. We believe that the convergence plots will not change in a significant manner if we consider DEL instead of standard deviation.

***Q 3.18*** *Table 4: The label should explicitly state that these are R2 results.*

**Reply**: The title of Table 5 is updated with additional text stating these are $R^2$ results.

***Q 3.19*** *L407: The description of the reference local distribution should be given in the methodology, i.e. table 3. and not here.*

**Reply**: The number of seeds and samples used for the analysis have now been mentioned in Section 3, page 18, line 693.

***Q 3.20*** *L418: The conclusion of the causes of the low performance at low TI should be illustrated by showing the local distributions, as you did in figures 11 or 12.*

**Reply**: Figure 16 has been added to the manuscript with conditional pdf plots corresponding to the poor prediction regions at low turbulence intensity (4%) values in Figure 15a.